# Understanding Likelihood Over-optimisation in Direct Alignment Algorithms

## Abstract

Direct Alignment Algorithms (DAAs), such as Direct Preference Optimisation (DPO) and Identity Preference Optimisation (IPO), have emerged as alternatives to online Reinforcement Learning from Human Feedback (RLHF) algorithms such as Proximal Policy Optimisation (PPO) for aligning language models to human preferences, without the need for explicit reward modelling. These methods generally aim to increase the likelihood of generating better (preferred) completions while discouraging worse (non-preferred) ones, while staying close to the original model's behaviour. In this work, we explore the relationship between completion likelihood and model performance in state-of-the-art DAAs, and identify a critical issue of likelihood over-optimisation. Contrary to expectations, we find that higher likelihood of better completions and larger margins between better and worse completion likelihoods do not necessarily lead to better performance, and may even degrade it. Our analysis reveals that while higher likelihood correlates with better memorisation of factual knowledge patterns, a slightly lower completion likelihood tends to improve output diversity, thus leading to better generalisation to unseen scenarios. Moreover, we identify two key indicators that signal when over-optimised output diversity begins to harm performance: *Decreasing Entropy over Top-k Tokens* and *Diminishing Top-k Probability Mass*. Our experimental results validate that these indicators are reliable signs of declining performance under different regularisation schemes, helping prevent over-optimisation and improve alignment with human preferences.

## 1 Introduction

Recent advancements in Large Language Models (LLMs) (Touvron et al., 2023; Achiam et al., 2023; Roziere et al., 2023; Dubey et al., 2024; Land & Bartolo, 2024) have significantly expanded their capabilities, enabling applications such as code generation, tool use, and interactive communication. As LLMs become increasingly powerful, the challenge of aligning them with human preferences has grown in importance. Direct Alignment Algorithms (DAAs), such as Direct Preference Optimisation (DPO) (Rafailov et al., 2023) and Identity Preference Optimisation (IPO) (Azar et al., 2024), have emerged as alternatives to Reinforcement Learning from Human Feedback (RLHF) (Ziegler et al., 2019; Bai et al., 2022) for training LMs on human preference data. These methods aim to bypass the traditional RLHF pipeline by directly optimising the policy without explicit reward modelling.

DAAs are designed to increase the likelihood of better completions while reducing the likelihood of worse ones, all while staying close to the original model's behaviour. However, a known issue with standard DAAs is that they may decrease the likelihood of better completions as long as the relative probability between better and worse completions increases (Rafailov et al., 2023; Pal et al., 2024). Recent research has sought to address this by focusing on maintaining a high likelihood for better completions (Pal et al., 2024). For example, several works (Pang et al., 2024; Hong et al., 2024), including Llama-3.1 (Dubey et al., 2024) and Nvidia Nemotron (Adler et al., 2024), introduce a scaled negative log-likelihood (NLL) loss on better completions, aiming to stabilise DAA training by preserving the desired formatting and preventing a drop in log probability for better completions. Despite these efforts, key research questions remain: *Is it truly necessary to maintain a higher likelihood of better completions, and aim for a larger likelihood margin between better and worse completions?* And if not, *How can we strike a balance for completion likelihood to maximise model performance in terms of alignment with human preferences?*

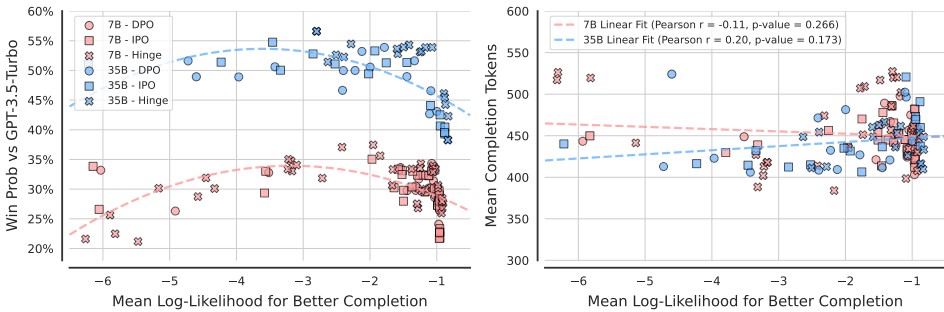

Figure 1: **Mean Log Likelihood (LLH) of Better Completion vs Win Probability (Left) and Average Number of Tokens in Model Outputs (Right).** We report 7B and 35B model results on the ULTRAFEEDBACK dataset. Our results indicate that: (1) A higher likelihood for better completions does not necessarily translate to higher win probability; and (2) There is no obvious correlation between the average number of tokens in model outputs and the likelihood of better completions.

In this work, we first explore the relationship between completion log-likelihood and model performance in state-of-the-art DAAs (§3). Specifically, we find that neither a higher likelihood of preferred completions nor larger margins between better and worse completion likelihoods necessarily lead to better performance (measured by win probability) and may even degrade it (§4.2), as shown in Figure 1. Furthermore, our experiments demonstrate that optimising both factors simultaneously also does not guarantee improvement. Our results reveal that while a higher likelihood of better completion generally has better memorisation of factual knowledge patterns, an excessively high likelihood can result in over-optimisation. In contrast, slightly lower completion likelihood tends to improve output diversity, thus leading to better generalisation to unseen scenarios (§4.3).

While avoiding an overly high completion likelihood tends to improve model diversity and generalisation, it is crucial to strike a balance between diversity and maintaining a high likelihood for desired outputs preferred by humans. To this end, our study outlines two key indicators that signal when overly generating diverse outputs begins to negatively impact model performance (§4.4): (1) **Decreasing Entropy over Top-$k$ Tokens**[1]: As the likelihood of better completions decreases during training, an increasing entropy suggests that tokens within better completions still have higher probabilities relative to other tokens in the Top-$k$, though the gap is narrowing. However, a decreasing entropy over the Top-$k$ tokens is a warning sign that the model is assigning disproportionately low probabilities to tokens within better completions, allowing other tokens to rise in probability, which may lead to outputs that are not aligned with human preferences. Notably, a reversed entropy trend is a particularly strong indicator of over-optimised diversity; and (2) **Diminishing Top-$k$ Token Probability Mass**: This occurs when the probability mass concentrated on the top $k$ most likely tokens declines, resulting in more random outputs and a higher likelihood of selecting tokens outside the top $k$. Such a flattening of the probability distribution can lead to phenomena such as code-switching (Doğruöz et al., 2021; Marchisio et al., 2024), making the model more prone to confusion. Our experimental results validate that these two indicators are strong predictors of declining model performance, providing critical markers to help avoid over-optimization while balancing diversity.

## 2 RELATED WORK

**Preference learning.** Recent years have seen significant progress in aligning LLMs with human preferences (Hosking et al., 2024; Kirk et al., 2024a). RLHF, pioneered by Christiano et al. (2017); Ziegler et al. (2019) and developed in subsequent works (Stiennon et al., 2020; Bai et al., 2022; Ouyang et al., 2022b), typically consists of three stages: supervised fine-tuning (SFT), reward modelling, and RL fine-tuning (Schulman et al., 2017; Mnih, 2016; Aryabumi et al., 2024; Ahmadian et al., 2024). The reward model is trained to predict human preferences between pairs of model outputs, while the RL phase optimises the model to maximise the reward (Ye et al., 2024; Lambert et al., 2024; Zhou et al., 2024a; Liu et al., 2024b). More recently, researchers have proposed Direct Alignment Algorithms (Rafailov et al., 2023; Zhao et al., 2023; Azar et al., 2024) that aim to simplify RLHF by directly optimising the policy without a reward modelling or RL phase.

---

[1]In this work, entropy measures uncertainty in token distribution, with a uniform distribution giving the highest entropy of 1 (maximum diversity) and a single-token distribution yielding 0 (no uncertainty).

**Over-optimisation for preference learning.** Over-optimisation occurs when a model's performance on a proxy measure improves while its true performance declines. Gao et al. (2023) was the first to extensively characterise this issue for RLHF, where optimisation against a learned reward model leads to increased proxy rewards, while actual task performance plateaus or worsens, a phenomenon termed "*reward over-optimisation*". Subsequent studies have observed similar patterns (Eisenstein et al., 2023; Touvron et al., 2023; Dubois et al., 2023). To mitigate this, researchers have proposed various approaches, such as using ensembles or data smoothing for reward modelling (Eisenstein et al., 2023; Zhang et al., 2024; Coste et al., 2024; Zhu et al., 2024; Yang et al., 2024b), and leveraging uncertainty signals (Yang et al., 2023; Zhai et al., 2023; Zhou et al., 2024b; Yang et al., 2024a). Rafailov et al. (2024) extended this analysis to DAAs, showing that even without an explicit reward model, DAAs exhibit similar over-optimisation patterns at higher KL-divergence budgets, where KL divergence as a primary metric. In contrast, we explore the DAAs' over-optimisation in the context of completion likelihood, which does not directly correlate with KL-divergence. Both increases and decreases in completion likelihood can result in higher KL divergence from the reference model. KL divergence is more about how far the model should move, while our likelihood analysis is more about which direction the model should move.

**Generalisation and diversity.** Generalisation and diversity in LM outputs has been a growing concern in the field of NLP, particularly regarding the impact of fine-tuning methods (Hendrycks et al., 2020). Several studies have explored how RLHF influences output diversity and generalisation. Khalifa et al. (2021); Perez et al. (2022) suggests that RLHF tends to produce models with reduced output diversity. Kirk et al. (2024b) highlights a trade-off between generalisation and diversity in current LLM fine-tuning, with RLHF showing better out-of-distribution generalisation but substantially decreased output diversity compared to SFT. This trade-off between alignment, performance, and diversity relates to the broader concept of "*alignment tax*" in LM fine-tuning. Bai et al. (2022); Ouyang et al. (2022a); Bai et al. (2023); Kotha et al. (2023) observed that aligning models with human preferences, through RLHF, can sometimes degrade performance on specific tasks, especially for smaller models. Various approaches have been proposed to mitigate the alignment tax (Noukhovitch et al., 2023; Shi & Lipani, 2024; Qi et al., 2024). For example, Ouyang et al. (2022a) suggested incorporating pretraining data into RLHF fine-tuning to minimise performance regressions on standard NLP datasets. However, these studies have not explored how the optimisation of completion likelihood correlates with model performance, including diversity and generalisation.

## 3 PRELIMINARIES

### 3.1 DIRECT ALIGNMENT ALGORITHMS

Direct Alignment Algorithms (DAAs) are a family of methods designed to train LMs to align with human preferences without the need for explicit reward modelling. These algorithms aim to optimise a policy model to maximise the probability of better completions over worse ones.

**Direct Preference Optimisation.** Direct Preference Optimisation (DPO) (Rafailov et al., 2023) is a foundational DAA method. The DPO loss function is defined as follows:

$$L_{\text{DPO}}(\pi_\theta; \pi_{\text{ref}}) = -\mathbb{E}_{(x, y_w, y_l) \sim D} \left[ \log \sigma \left( \beta \Delta(x, y_w, y_l) \right) \right], \tag{1}$$

$$\Delta(x, y_w, y_l) = \log \frac{\pi_\theta(y_w|x)}{\pi_{\text{ref}}(y_w|x)} - \log \frac{\pi_\theta(y_l|x)}{\pi_{\text{ref}}(y_l|x)}, \tag{2}$$

where $\pi_\theta$ is the policy model being optimised, $\pi_{\text{ref}}$ is a reference model where $\pi_\theta$ is initialised from, $D$ is the dataset of preference pairs, $x$ is the input, $y_w$ and $y_l$ are the better and worse completions respectively, $\sigma$ is the sigmoid function, and $\beta$ is a temperature hyperparameter. The term $\Delta(x, y_w, y_l)$ quantifies the difference in log probabilities between better and worse completions.

**Identity Preference Optimisation.** Identity Preference Optimisation (IPO) (Azar et al., 2024) is a variant of DAA methods. Specifically, IPO uses a quadratic loss function, which is defined as:

$$L_{\text{IPO}}(\pi_\theta; \pi_{\text{ref}}) = \mathbb{E}_{(x, y_w, y_l) \sim D} \left[ \left( \tau \Delta(x, y_w, y_l) - \frac{1}{2} \right)^2 \right], \tag{3}$$

where $\tau$ is a temperature hyperparameter. This formulation aims to push the difference in log probabilities $\Delta(x, y_w, y_l)$, defined within the DPO framework, towards a target value of $\frac{1}{2\tau}$.

**Hinge Loss.** The hinge loss method (Zhao et al., 2023; Liu et al., 2024a) represents another variation within the DAA framework. Specifically, we adopt the loss function from SLiC-HF (Zhao et al., 2023), which is defined as follows:

$$L_{\text{Hinge}}(\pi_\theta; \pi_{\text{ref}}) = \mathbb{E}_{(x, y_w, y_l) \sim D} \left[ \max \left( 0, \gamma - \log \frac{\pi_\theta(y_w|x)}{\pi_\theta(y_l|x)} \right) \right], \qquad (4)$$

where $\gamma$ is a hyperparameter and we set to $\gamma = 1$ for simplicity. In line with Zhao et al. (2023), we incorporate a regularisation term into the hinge loss, defined as follows:

$$L_{\text{reg}}(\pi_\theta; \pi_{\text{ref}}) = \mathbb{E}_{(x, y_w, y_l) \sim D} \left[ \log \left( 1 + \exp \left( 1 - \log \left( \frac{\pi_\theta(y_w|x)}{\pi_{\text{ref}}(y_w|x)} \right) \right) \right) \right], \qquad (5)$$

which represents a smoothed version of hinge loss (Huber, 1992; Cristianini & Shawe-Taylor, 2000). This term encourages the likelihood of better completions to remain higher than that of the reference model. The total hinge loss is given by $L_{\text{Hinge}}(\pi_\theta; \pi_{\text{ref}}) = L_{\text{Hinge}}(\pi_\theta; \pi_{\text{ref}}) + \alpha L_{\text{reg}}(\pi_\theta; \pi_{\text{ref}})$, where $\alpha$ is a scaling coefficient.

## 3.2 BETTER LIKELIHOOD SUPPORT

Standard DAAs do not guarantee an increase in the absolute probability of better completions. This can lead to scenarios where the model assigns very low probabilities to both better and worse completions, as long as the better completion has a higher relative probability.

**Negative Log-Likelihood Loss.** To mitigate this issue, Negative Log-Likelihood (NLL) loss is commonly employed as a regularisation term in DAA (Hong et al., 2024; Pang et al., 2024; Adler et al., 2024; Dubey et al., 2024). It encourages the policy to maintain a high likelihood of better completions. The NLL loss is formulated as:

$$L_{\text{NLL}}(\pi_\theta) = -\mathbb{E}_{(x, y_w) \sim D} \left[ \log \pi_\theta(y_w|x) \right], \qquad (6)$$

where $y_w$ represents the better completion for a given input $x$. This loss term is typically combined with the primary objective of the DAA using a scaling coefficient $\lambda$.

Several other regularisation methods have been proposed to address this issue. For example, Pal et al. (2024) introduces an additional term, $- \max \left( 0, \log \frac{\pi_\theta(y_w|x)}{\pi_\theta(y_l|x)} \right)$, to $\Delta(x, y_w, y_l)$ to ensure that the log-likelihood of better examples remains high relative to that of the reference model. In this work, we mainly discuss the impact of Negative Log-Likelihood Loss.

## 4 UNDERSTANDING THE IMPACT OF COMPLETION LIKELIHOOD

### 4.1 EXPERIMENTAL SETUP

**Model and Datasets.** In our experiments, we utilise two instruction-tuned models: Cohere Command R (7B) and Cohere Command R (35B) (Cohere For AI, 2024). We train and evaluate them on two datasets: (1) A binarised version of ULTRAFEEDBACK (Tunstall et al., 2024), which is collected based on Cui et al. (2024), containing 62,600 training examples and 647 examples for evaluation. (2) A Binarised preference dataset BINARIZEDPREF, which comprises over 100,000 examples (see details in Appendix §A). These include annotated conversational data across multiple languages, synthetic code generation, and specialised tasks such as length control, safety, tool use, and natural language-to-SQL generation.

**Training and Evaluation Details.** For each method (Hinge, DPO, and IPO), we test six different values for its hyper-parameter (*i.e.,* $\alpha$, $\beta$, or $\tau$), respectively. We use a batch size of 32 for both training and evaluation, with a maximum sequence length of 8192. The model is trained with a peak learning rate of either $5 \times 10^{-6}$ or $1 \times 10^{-5}$ and an end learning rate ratio of 0.1. Following recent studies (Ouyang et al., 2022a; Howard & Whitaker, 2023; Shi et al., 2024), we train all models

within a single epoch. The learning rate warms up over 128 steps. We monitor the model training every 50 steps to apply early stopping. We use the Adam optimiser (Kingma, 2014) with $\beta_1 = 0.9$, $\beta_2 = 0.95$, $\epsilon = 1 \times 10^{-8}$, an additive weight decay of 0.1, and a gradient clipping norm of 1.0. The model training is conducted on TPU v5-128 for the 7B model and TPU v5-256 for the 35B model, utilising the flash attention (Dao et al., 2022) to improve training efficiency. For both DPO and IPO, we use the sum of the token log-likelihoods as the completion log-likelihood during training. For the Hinge method, we compute the average token log-likelihood instead for better performance. During evaluation, we calculate the log-likelihood for both the better and worse completions from the validation set. For all methods, we report the average of token log-likelihoods for better and worse completions respectively, without normalising against the reference model. Additionally, we monitor the difference in log-likelihood between better and worse completions.

**Generalisation Evaluation.** Following the previous work (Kirk et al., 2024b), we evaluate the model in open-ended text generation tasks to assess generalisation ability. Specifically, we employ the LLM-as-a-Judge framework (Zheng et al., 2023; Taori et al., 2023) with a reward model to compare our models' outputs against leading models, including GPT-3.5-Turbo, GPT-4o (Achiam et al., 2023), Claude-3-Sonnet (Claude, 2024), Llama-3 8B and 70B Chat (Dubey et al., 2024). The evaluation uses a closed-source reward model, which ranked the top position on REWARDBENCH (Lambert et al., 2024), validating that the evaluation provides a reliable proxy for human preferences. We use win probability, denoted as $P_{\text{win}}$, as the primary evaluation metric. It is computed as:

$$P_{\text{win}} = \sigma(r_v - r_c), \tag{7}$$

where $\sigma(\cdot)$ is the sigmoid function, $r_v$ is the reward assigned to the policy model's output, and $r_c$ is the reward assigned to the competitor model's output by the same reward model. We prompt models with 433 diverse prompts, including code generation, chain-of-reasoning questions, closed QA, and length control (see Appendix A for examples and details). During the decoding, we use a top-$p$ probability threshold of $p = 0.75$, a temperature of 0.5, and a maximum limit of 2048 tokens.

**Diversity Evaluation.** To assess output diversity, we also measure **Per-Input Diversity**, defined as the average diversity of the output sets over inputs, and **Cross-Input Diversity**, which captures the diversity of outputs across different inputs, similar to previous works (Kirk et al., 2024b; Hong et al., 2024). However, instead of generating a set of $K$ outputs from the model, we take a more efficient way to measure Per-Input Diversity. Specifically, we compute the entropy over the top $k$ tokens with the highest probability in the model's next token distribution (Kuhn et al., 2023). Let $p_k$ represent the probability distribution over the top $k$ tokens, and $H(p_k)$ represent the entropy of the distribution. The entropy is calculated using the following formula:

$$H(p_k) = -\sum_{i=1}^{n} p_i \log_b(p_i), \tag{8}$$

where $b$ is the logarithm base. Here we set $b = 2$ and $k = 10$. This formula quantifies the uncertainty within the top $k$ token predictions as a proxy for Per-Input Diversity. This entropy is highest when the output is minimally informative: predicting the same probability for all possible tokens, indicating more diverse outputs. To evaluate Cross-Input Diversity, we use distinct N-grams (Li et al., 2016), which counts the unique N-grams across model outputs and averages them over $n = 1, 2, 3, 4, 5$. Following Kirk et al. (2024b), we use the expectation-adjusted distinct N-grams (EAD) formula to remove the bias towards shorter outputs.

**Factuality Evaluation.** We also evaluate model factuality performance on open-domain question-answering tasks using NATURALQUESTIONSOPEN (Kwiatkowski et al., 2019) and TRIVIAQA (Joshi et al., 2017) validation sets, with 3610 and 7993 examples respectively. Greedy decoding is used to ensure deterministic outputs, and the word-level $F_1$ score is reported.

### 4.2 EVALUATING LIKELIHOOD OVER-OPTIMISATION

In this section, we explore the relationship between model likelihood and performance. Below, we discuss our key findings in detail.

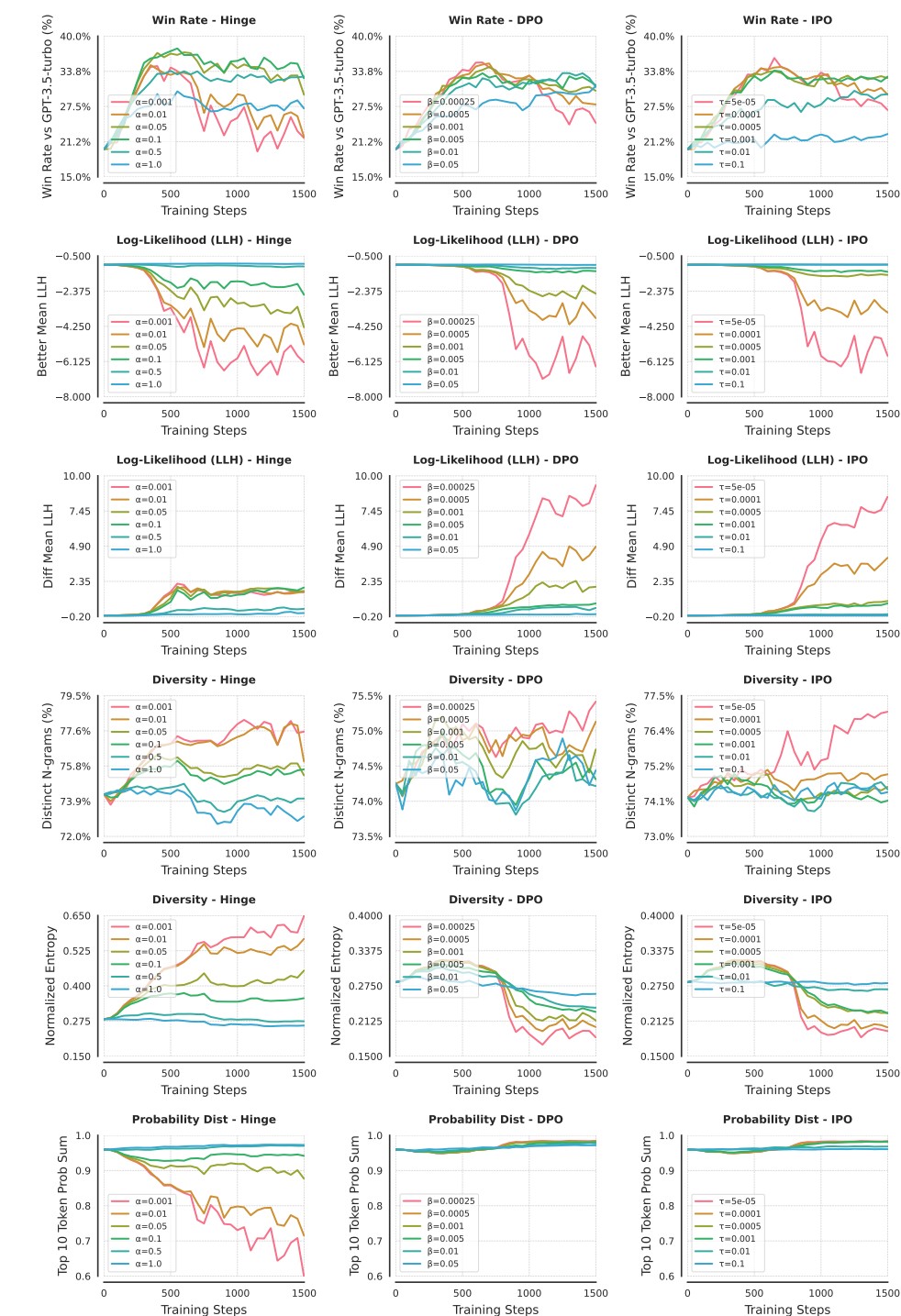

Figure 2: **Learning curves across training steps for various metrics**. Results are reported for the 7B models using the Hinge, DPO, and IPO on the ULTRAFEEDBACK dataset. Our results indicate that: (1) A higher likelihood for better completions does not necessarily improve model performance. (2) Lower Completion likelihood improves the models' Cross-Input Diversity. (3) Decreasing in Probability Mass in Top k Tokens and Decreasing Entropy over Top-k tokens are signals for likelihood over-optimisation.

**1) Higher likelihood for better completions and larger gaps between better and worse completions do not necessarily improve model performance.** As shown in Figure 1, we plot the likelihood of better completions against the win probability (compared to GPT-3.5-Turbo) with dif-

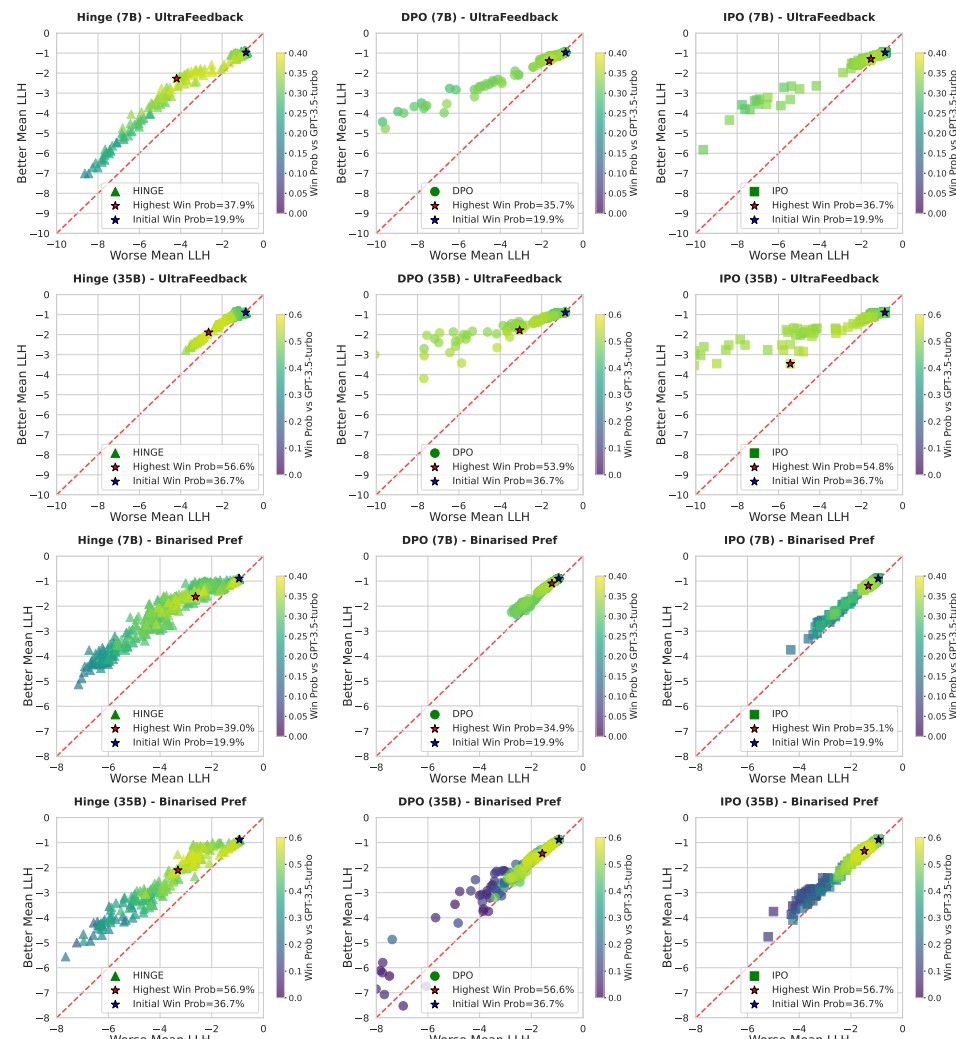

Figure 3: **Win Probability Heatmaps Across Better and Worse Mean Log-Likelihoods.** Results are reported for both 7B and 35B models on ULTRAFEEDBACK and BINARIZEDPREF datasets. Best performance does not always occur at the Pareto frontier of high likelihood for better completions and low likelihood for worse completions.

ferent methods across two model sizes, with points recorded every 500 steps. Our analysis reveals that simply increasing the likelihood of better completions does not consistently result in performance improvements. Previous work in classical RLHF has established scaling laws for reward model scores (Gao et al., 2023). Similarly, Figure 1 exhibits a clear scaling law behaviour. We extend their analysis to the relationship between win probability and the log-likelihood of better completions in DAAs. When fitting the data to a second-degree polynomial, the Root Mean Square Error decreases by approximately 24.42% for the 7B model and 25.78% for the 35B model, compared to a linear fit. We show similar results when comparing against different models, including GPT-4o, Claude-3-Sonnet, Llama-3-8B, and Llama-3-70B-Chat, in Figure 7 of Appendix §B.

Figure 2 tracks win probability alongside the average log-likelihood difference between better and worse completions throughout training. Notably, while larger differences in log-likelihood, such as those represented by the pink line typically with the largest difference, are often observed, they do not correspond to better performance. Instead, excessively larger likelihood gaps can lead to performance degradation in win probability, especially for DPO and IPO after 1,000 steps. We observe similar results for the 35B model on BINARIZEDPREF using Hinge, DPO, and IPO in Appendix §B.

Figure 3 presents a heatmap of win probabilities based on the better and worse completion log-likelihoods on ULTRAFEEDBACK and BINARIZEDPREF datasets, using both 7B and 35B models.

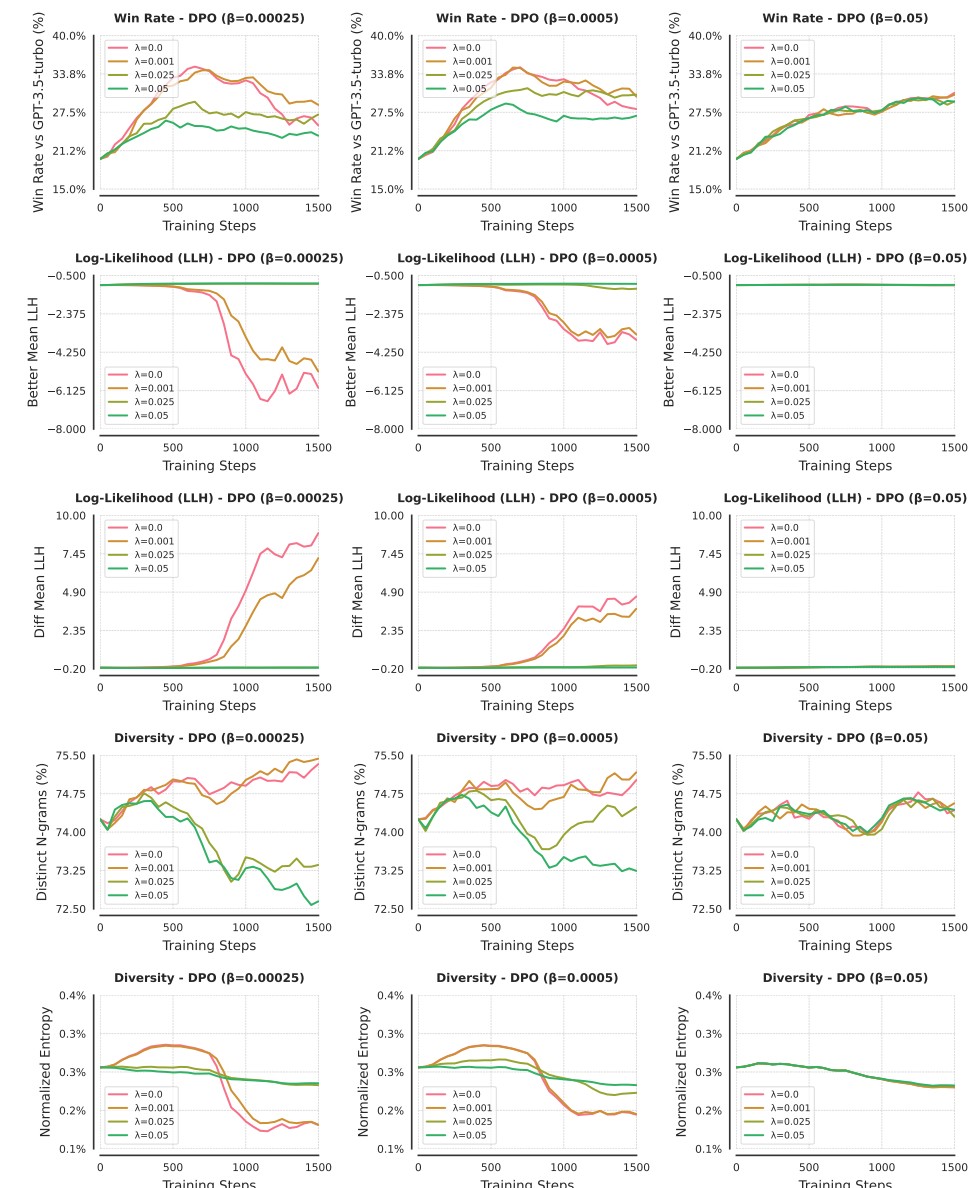

Figure 4: **Learning curves for DPO with different weights ($\lambda$) of NLL loss**. We report the performance with different values of $\beta$ and $\lambda$ on the ULTRAFEEDBACK dataset. Our results indicate that: (1) Training Negative Log-Likelihood Loss on better completions has limited influence on the model when it cannot affect completion likelihood. (2) A reversed entropy trend trending for entropy is a strong indicator of diversity over-optimisation.

Points are plotted every 50 steps. Our findings indicate that the best performance (highlighted by the red star) does not occur at the Pareto frontier of maximising the likelihood of better completions while minimising it for worse ones. Instead, optimal performance is often found in the middle range.

**2) Length Correlation.** We investigate the relationship between the mean log-likelihood of better completions and the average number of tokens in completions, as shown in Figure 1. To quantify this relationship, we calculate the Pearson correlation coefficient and perform its associated significance test. The null hypothesis posits no linear relationship between these two variables. For the 7B model, we find a weak negative correlation ($r = -0.114$, $p$-value = 0.266), while the 35B model shows a weak positive correlation ($r = 0.198$, $p$-value = 0.173). In both cases, the p-values exceed the conventional significance level of 0.05, indicating insufficient evidence to reject the null hypothesis.

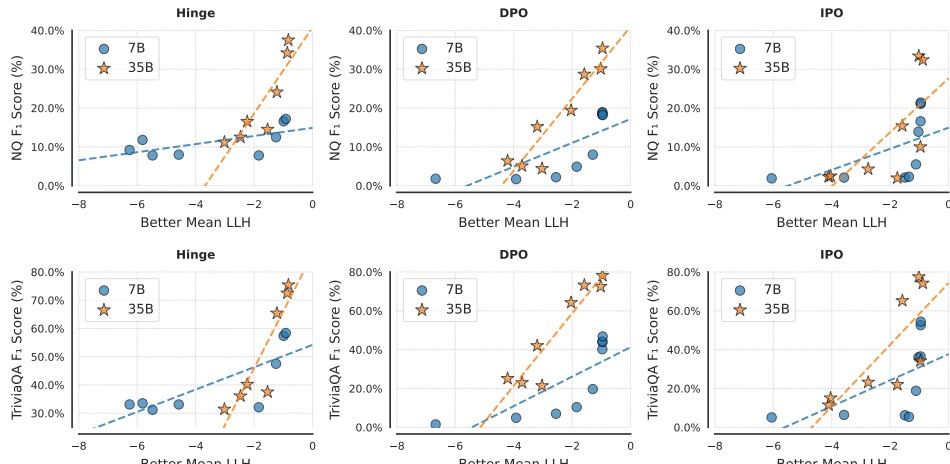

Figure 5: NATURALQUESTIONSOPEN and TRIVIAQA vs Better Mean LLH on the ULTRAFEED-BACK dataset. A higher LLH tends to memorise the factuality knowledge better.

**3) Training Negative Log-Likelihood Loss on better completions has limited influence on the model when it cannot affect completion likelihood.** As shown in Figure 4, we experiment with DPO using three different values of $\beta$, adding NLL loss as an auxiliary loss with four $\lambda$ coefficients. Our results indicate that when there is limited impact on the likelihood (from the left column to the right column), the NLL loss has minimal impact on model performance. This suggests that NLL loss can be seen as a tool to regulate completion likelihood, but it remains susceptible to likelihood over-optimisation: higher likelihood may lead to a sub-optimal performance. We observe similar results on BINARIZEDPREF using the 35B model, as shown in Figure 11 of Appendix §B.

### 4.3 GENERALISATION AND DIVERSITY

In this section, we explore the impact of model likelihood on generalisation and diversity.

**1) Lower Completion likelihood improves the models' Cross-Input Diversity.** Figure 2 presents Cross-Input Diversity (measured by distinct N-grams) of the model outputs throughout training. Specifically, within each DAA, models with lower likelihood tend to produce more diverse outputs. For example, the pink lines for DAAs indicate that models with lower completion likelihood typically show the highest level of Cross-Input Diversity scores throughout training. Better output diversity tends to improve their generalisation to unseen scenarios, as reflected in increased win probability at the early stage of the training phase. Figure 4 further demonstrates that output diversity follows a similar trend under the different regularisation (*i.e.,* Negative Log-Likelihood Loss), suggesting a strong correlation between likelihood and model diversity. However, it is worth noting that the relationship between diversity and win probability is not linear. While some diversity is beneficial for generalisation, excessive diversity can lead to performance degradation, similar to our previous discussion in §4.2. We will explore this phenomenon further in §4.4.

**2) Higher Likelihood tends to have better memorisation of factual patterns.** Figure 5 showcases the relationship between model performance on NATURALQUESTIONSOPEN and TRIVIAQA and the log-likelihood of better completions. Our findings reveal a clear trend: higher mean log-likelihood values are associated with improved $F_1$ scores. A higher $F_1$ reflects better memorisation for some specific patterns, which can come at the expense of diversity. This can create a trade-off between the ability to recall facts and the capacity to generate diverse, adaptive outputs in more creative or open-ended tasks. To understand the potential issue of stylistic variations in answers, we provide a further analysis with case studies and LLM-as-a-Judge as evaluation in Appendix §C. Specifically, instead of relying on exact string matching, which can be overly rigid, we employ an LLM-as-a-Judge using the GPT-4o model. Our analysis reveals that while the model performance from LLM-as-a-Judge evaluation consistently yields higher performance metrics, it demonstrates a trend similar to the $F_1$ score.

## 4.4 Signals for Likelihood Over-optimisation

We have shown that completion likelihood correlates with model performance due to increased output diversity. However, the key question remains: when should we stop reducing completion likelihood? Here, we outline two indicators of over-optimising likelihood.

**1) Decreasing Entropy over Top-$k$ tokens (Per-Input Diversity).** Figure 2 and 4 presents Per-Input Diversity (measured by the entropy) of the model outputs throughout training. For DPO and IPO curves, at the beginning of the training, the Per-Input Diversity increases, signifying a broader distribution of selected tokens and a more uniform output distribution for the next token prediction. Considering that the better completion likelihood is decreasing across the training, the increase of entropy at the beginning phase indicates that those tokens from better completion have a higher probability at the initial policy model over other tokens in the top $k$ (here $k = 10$). The decrease better completion likelihood gives the model a better chance to select other tokens, which increases diversity and enhances generalisation, as reflected in the win probability. However, at a certain point in training, this trend reverses. As Per-Input Diversity (entropy) starts decreasing, the model begins to over-prioritise certain tokens. This suggests that those tokens in the better completion now have an overly low likelihood, lower than other tokens in the top $k$. Despite this, Cross-Input Diversity keeps increasing, which indicates that the model is still generating diverse outputs, but now it includes tokens that are less relevant or nonsensical, *i.e.,* tokens that humans do not prefer. Notably, the turning points of entropy often coincide with those of win probability for DPO and IPO, as the model's outputs become less aligned with desirable outcomes.

**2) Decreasing in Probability Mass in Top $k$ Tokens.** In another scenario, the entropy of the top 10 tokens continues to increase, suggesting a progressively broader and more uniform output distribution (refer to the hinge curves in Figure 2). This suggests that even as the likelihood of better completions decreases, the model does not tend to over-prioritise any specific tokens during training. However, this can result in degraded model performance. As depicted in the bottom row of the figure, the probability mass of all top-10 tokens diminishes, leading to more random outputs, with an increased likelihood of selecting tokens outside the top 10. This can introduce issues such as code-switching, where the model becomes prone to world-level language confusion when the number of tokens in the sampling nucleus is high and the distribution becomes too flat (Doğruöz et al., 2021; Marchisio et al., 2024). Interestingly, hinge loss models do not exhibit the same patterns observed with DPO and IPO. This could be attributed to the fact that DPO and IPO apply different forms of regularisation compared to hinge loss.

To demonstrate the generalisability of our findings, we provide additional experimental on different datasets with different model sizes in Figure 8, 9, and 10 of Appendix §B.

## 5 Epilogue

**Limitations.** This study primarily focuses on two models (7B and 35B), which may not fully represent the broader spectrum of LLMs available. However, most LLMs are very standard transformers (Vaswani et al., 2017), and we would not expect other LLMs to behave differently. While we acknowledge the reviewer's concern about testing additional methods such as KTO Ethayarajh et al. (2024) or ORPO (Hong et al., 2024), our experiments with major DAA families (*e.g., ,* DPO, IPO, SLiC) provide strong evidence for the generalisability of our findings, which we leave for future work to validate further.

**Implications for Practical Applications.** The findings of this study have several implications for enhancing offline preference learning methods in practical applications: (1) ***Early stopping signal***. In practice, we can integrate entropy/probability mass monitoring into the training loop. Training can employ adaptive methods like early stopping once entropy falls below a specific threshold. (2) ***Adaptive regularisation for over-optimisation***. Rather than using a fixed coefficient for the NLL loss (Dubey et al., 2024), we could implement an adaptive regularisation based on the entropy and probability mass, *i.e.,* adding dropout or noise to prevent over-prioritisation of tokens or adding an explicit regularisation term that maintains a certain degree of entropy and the probability mass of the top-$k$ tokens. While maintaining a certain degree of entropy and probability mass of the top-$k$ tokens is important, care should be taken not to overly constrain the model, as some tasks inherently require a broader token distribution (*e.g.,* give me a random number between 0 and 10).

## REPRODUCIBILITY STATEMENT

To ensure the reproducibility of our results, we have taken comprehensive steps to provide detailed information about our experimental setup. In Section 4.1, we offer full details on the models used (7B and 35B parameter models) and the datasets (ULTRAFEEDBACK and BINARIZEDPREF), including exact versions and sizes. While the 7B model and reward model are closed-source, and the 433 prompts for the LLM-as-a-Judge framework are proprietary, we provide a summary of the prompt dataset to give insight into its composition. All hyperparameters for training, including learning rates, batch sizes, and optimizer settings, are specified. We detail the hardware used (TPU v5-128/256) and provide comprehensive descriptions of all evaluation metrics. Statistical analyses, including Pearson correlation coefficients and p-values, are reported in Section 4.2. The ULTRA-FEEDBACK dataset is publicly available, and while BINARIZEDPREF is proprietary, we describe its contents and size. Importantly, we test our findings on ULTRAFEEDBACK, which is a public dataset, indicating that our findings are generalisable. While some aspects could not be fully open-sourced due to the use of proprietary models or data, we have described these in as much detail as possible. Furthermore, we posit that our findings are likely generalisable to other LLMs, as most LLMs (*e.g.,* Llama, Gemini) are based on standard transformer architectures (Vaswani et al., 2017). For example, the Llama model family has very standard features such as RoPE embeddings (Su et al., 2024). Indeed, the designers note that they tried to avoid innovating on the model architecture (Dubey et al., 2024). As such, we would not expect significantly different behaviours. We welcome questions from the community and are committed to providing additional clarification.

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

APPENDIX OVERVIEW

The appendix is structured as follows:

**Appendix §A**    provides a detailed description of evaluation datasets, including examples and statistical summaries.

**Appendix §B**    presents supplementary experimental results, including analyses of win probability, likelihood scaling, and the effects of different regularization techniques.

**Appendix §C**    further investigates model performance on NATURALQUESTIONSOPEN and TRIVIAQA.

## A   DATASETS

This section provides an in-depth look at the datasets used in our experiments, focusing on the BINARIZEDPREF, LLM-as-a-Judge framework, NATURALQUESTIONSOPEN, and TRIVIAQA datasets.

**BINARIZEDPREF Dataset.**    The BINARIZEDPREF collection process used a robust multi-source approach combining professional annotators, multiple independent annotation pipelines, and various validation methods. The foundation comes from professional annotation services ( 70% of data), with rigorous quality control through multi-annotator consensus, adversarial validation sets, and specialized verification datasets for issues like hallucination and repetition. We've ensured broad domain coverage, incorporating specialised modules for code generation, RAG interactions, STEM, and medical domains while maintaining strong multilingual capabilities across French, Spanish, Korean, Japanese, German, and Italian - including dedicated datasets for handling code-mixing and language transition cases. Quality control is implemented through multiple layers: consensus-based annotation (1-3 annotators depending on complexity), dedicated adversarial validation sets, and specific datasets targeting quality aspects like anti-repetition, length control, and format adherence. The data is predominantly recent (2024), with carefully weighted components and explicit test sets for key capabilities. We use strategic copy multipliers (up to 5x) for crucial capabilities, and the entire dataset is organised into functional groups (multilingual, code, RAG) to ensure balanced training across all target capabilities.

**LLM-as-a-Judge Framework Dataset.**    We utilize a diverse set of prompts for the LLM-as-a-Judge framework. Figure 1 illustrates a representative example from this dataset, showcasing different generations from various competitor models. To provide insight into the composition of our LLM-as-a-Judge dataset, Figure 6 presents the distribution of prompt examples. This visualisation helps to understand the variety and balance of the prompts used in our evaluation framework.

**NATURALQUESTIONSOPEN Dataset.**    Table 2 presents examples from the NATURALQUESTIONSOPEN dataset[2], showcasing the types of questions and answers used in our evaluation. The NATURALQUESTIONSOPEN dataset, introduced by Kwiatkowski et al. (2019), is an open-domain question-answering benchmark. It consists of English questions paired with possible answer strings, all answerable using English Wikipedia content. Each data instance contains a question field and an answer field with potential correct responses. We use the validation set for our evaluation. Table 2 presents representative examples from this dataset, illustrating the types of questions and answers used in our evaluation.

**TRIVIAQA Dataset.**    The TRIVIAQA dataset[3] is a comprehensive reading comprehension benchmark containing over 650,000 question-answer-evidence triples Joshi et al. (2017). It includes 95,000 question-answer pairs, each accompanied by an average of six independently gathered evidence documents. This structure provides high-quality distant supervision for question-answering

---

[2]https://huggingface.co/datasets/google-research-datasets/nq_open/viewer/nq_open/validation
[3]https://huggingface.co/datasets/mandarjoshi/trivia_qa/viewer/rc.wikipedia/validation

Table 1: Examples for datasets.

| Role | Example |
|------|---------|
| **User** | I am looking to put a house on the market. 4 bedrooms, 3 bathrooms, 3000 sq ft. Nice neighbourhood and local school, strong community feel. Safe for kids, lots of young families. Can you write me a general description for the house in around 100-150 words? Make it professional sounding, emphasizing key aspects that you think would be attractive to young families. |
| **GPT-3.5-turbo** | Welcome to this spacious and inviting 4 bedroom, 3 bathroom home located in a desirable neighbourhood with a strong sense of community. Boasting over 3000 sq ft of living space, this property is perfect for growing families looking for a safe and welcoming environment to call home. The open-concept layout offers ample room for entertaining, while the well-appointed bedrooms provide a peaceful retreat for all. The nearby local school and abundance of young families in the area make this the ideal setting for children to thrive and make lasting friendships. Don't miss out on the opportunity to make this house your family's forever home. |
| **GPT-4o** | Welcome to your dream family home! Nestled in a vibrant, family-friendly neighborhood, this spacious 4-bedroom, 3-bathroom residence offers 3,000 sq ft of comfortable living space. The open floor plan seamlessly connects the living, dining, and kitchen areas, making it perfect for both everyday living and entertaining. Each bedroom is generously sized, providing ample space for relaxation and privacy.\n\nLocated in a top-rated school district, this home ensures your children will receive a quality education. The community boasts a strong, supportive atmosphere with plenty of young families, making it easy to form lasting friendships. Safety is a priority here, with well-maintained streets and parks where kids can play freely. Don't miss the opportunity to create cherished memories in this exceptional home. Schedule a viewing today! |
| **Claude-3-Sonnet** | Nestled in the heart of a vibrant community, this spacious 3,000 square foot home offers the perfect blend of comfort and convenience for growing families. With four generously sized bedrooms and three beautifully appointed bathrooms, this residence provides ample space for daily living and entertaining. The neighborhood boasts a strong sense of community, with a local school nearby and a safe, family-friendly environment where children can thrive. The area is home to numerous parks, playgrounds, and recreational facilities, ensuring endless opportunities for outdoor activities and quality family time. Designed with modern amenities and a welcoming ambiance, this property is an exceptional choice for families seeking a comfortable and secure haven to call home. |
| **Llama-3-70b-chat-hf** | Here is a general description for your house:\n\nWelcome to this stunning 4 bedroom, 3 bathroom residence, perfectly situated in a highly sought-after neighborhood that embodies the essence of community living. With its spacious 3000 square feet of living space, this beautiful home offers the perfect blend of comfort, functionality, and charm. Ideal for young families, this property boasts a prime location within walking distance to a top-rated local school, ensuring a safe and convenient commute for kids. The neighborhood itself is a hub of activity, with a strong sense of community and a plethora of young families, making it the perfect environment for children to grow and thrive. With its generous layout, abundance of natural light, and exceptional curb appeal, this property is sure to impress even the most discerning buyers. Don't miss out on this incredible opportunity to make this house your dream home!\n\nLet me know if you'd like me to make any changes! |
| **Llama-3-8b-chat-hf** | Here is a general description for your house:\n\n"Welcome to this stunning 4-bedroom, 3-bathroom residence, perfectly situated in a highly sought-after neighborhood. This beautiful home offers a spacious 3000 square feet of living space, ideal for growing families. With its prime location, you'll enjoy the convenience of being close to top-rated local schools, making it an ideal choice for young families. The neighborhood exudes a strong sense of community, with plenty of opportunities for kids to make friends and for parents to connect with neighbors. The home itself is a haven of comfort and style, with ample natural light, modern fixtures, and plenty of storage. With its safe and family-friendly atmosphere, this property is the perfect place to call home. Don't miss this incredible opportunity to make memories with your loved ones in this wonderful community!\n\nLet me know if you'd like me to make any changes! |

Table 2: Examples for NATURALQUESTIONSOPEN.

| Question | Answer |
|----------|--------|
| who does the voice of mickey mouse on mickey mouse clubhouse? | ['Bret Iwan', 'Wayne Allwine'] |
| who wrote knock knock knocking on heavens door? | ['Bob Dylan'] |

Table 3: Examples for TRIVIAQA.

| Question | Answer |
|----------|--------|
| Who was the next British Prime Minister after Arthur Balfour?? | ['Sir Henry Campbell-Bannerman', 'Campbell-Bannerman', 'Campbell Bannerman', 'Sir Henry Campbell Bannerman', 'Henry Campbell Bannerman', 'Henry Campbell-Bannerman'] |
| Which Lloyd Webber musical premiered in the US on 10th December 1993?? | ['Sunset Blvd', 'West Sunset Boulevard', 'Sunset Boulevard', 'Sunset Bulevard', 'Sunset Blvd.'] |

tasks. However, we do not use any evidence in our experiments. We use the validation set for our evaluation. Table 3 presents representative examples from the TRIVIAQA dataset.

## B ADDITIONAL EXPERIMENTAL RESULTS

As supplementary of the main experiment, we provide the following experiments.

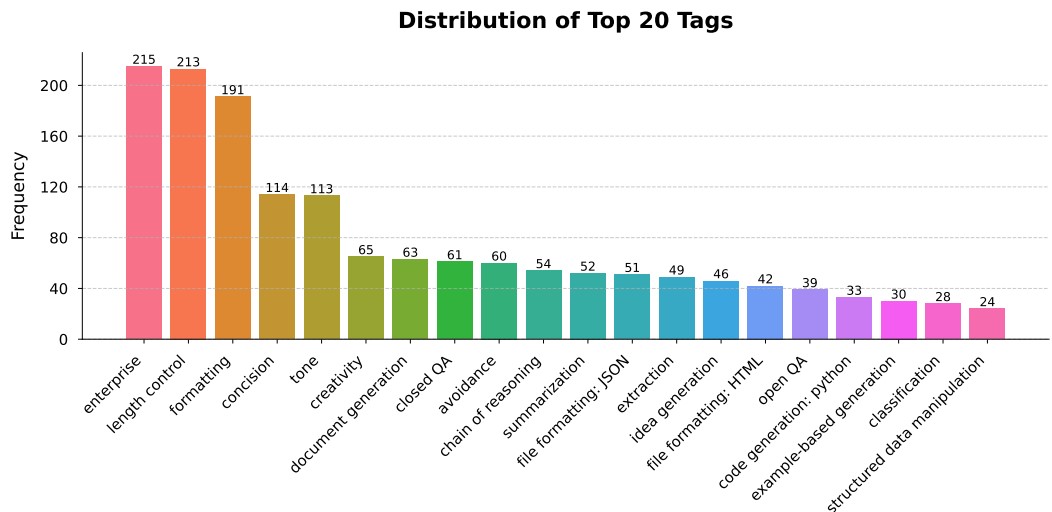

Figure 6: Distribution of LLM-as-the-judge prompt dataset.

**Win Probability vs. Better Completion Likelihood.** Figure 7 illustrates the relationship between win probability and better mean likelihood across different competitor models, including GPT-4, Claude-3-Sonnet, Llama-3-8B, and Llama-3-70B-Chat. We record points every 500 steps across varying hyperparameters for each method. Our results are consistent with our findings in the main text (§4.2), suggesting that simply increasing the likelihood of better completions does not consistently result in performance improvements.

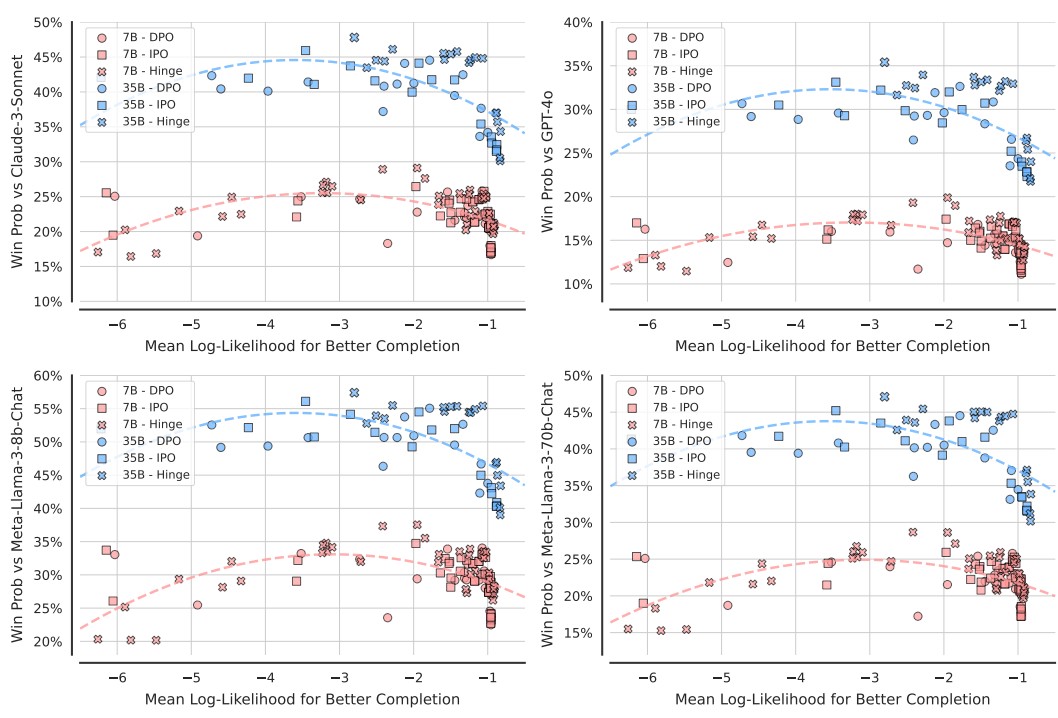

Figure 7: Win Probability vs Better Mean Likelihood Scaling Law. with different competitor models. including GPT-4o, Claude-3-Sonnet, Llama-3-8B, and Llama-3-70B-Chat

**IPO Learning curves with 7B model on the ULTRAFEEDBACK dataset.** To demonstrate the generalisability of our findings, we experiment with the IPO using three different values of $\tau$, adding NLL loss as an auxiliary loss with four $\lambda$ coefficients on the ULTRAFEEDBACK dataset using the 7B model. Figure 8 illustrates several key findings:

1. **Likelihood and Performance Correlation:** As shown in the first and second rows of the figure, a Higher likelihood for better completions and larger gaps between better and worse completions do not necessarily translate to improved model performance.

2. **Likelihood and Cross-Input Diversity:** Lower completion likelihood tends to enhance the models' Cross-Input Diversity, as shown in the second and fourth rows, where lower better completion likelihood generally corresponds to improved Cross-Input Diversity.

3. **Entropy and Over-optimisation:** Decreasing entropy over top-$k$ tokens (Per-Input Diversity) appears to be an indicator of over-optimisation for diversity. The fifth row demonstrates that curves with lower entropy typically do not perform as well, as reflected in their win probability. Particularly, this result shows that the turning points of the entropy, which transits from the increasing diversity to the decreasing entropy is a strong indicator of the over-optimisation for diversity.

4. **Probability Mass Distribution:** We do not observe a decrease in probability mass in top $k$ tokens in this case, as shown in the last row of the figure. This observation aligns with our findings: in runs without decreasing entropy, we do not observe a significant decline in win probability.

**Learning curves with 7B model on the BINARIZEDPREF dataset.** To demonstrate the generalisability of our findings, we perform additional experiments using the 7B model on the BINARIZEDPREF dataset. The results, consistent with our previous observations, underscore the broad applicability of our insights across various datasets. Figure 9 illustrates several key findings:

1. **Likelihood and Performance Correlation:** Higher likelihood for better completions and larger gaps between better and worse completions do not necessarily translate to improved model performance. This is evident in the first and second rows of the figure, where models with the highest better completion likelihood do not achieve the best performance.

2. **Likelihood and Cross-Input Diversity:** Lower completion likelihood tends to enhance the models' Cross-Input Diversity. This trend is observable when comparing the second and fourth rows, where lower better completion likelihood generally corresponds to improved Cross-Input Diversity.

3. **Entropy and Over-optimisation:** Decreasing entropy over top-$k$ tokens (Per-Input Diversity) appears to be a good indicator of over-optimisation for diversity. The fifth row demonstrates that curves with overly low entropy do not perform as well (*i.e.,* pink curves), as reflected in their win probabilities. Additionally, as the entropy begins to rise again, an improvement in win probability is also observed.

4. **Probability Mass Distribution:** We do not observe a decrease in probability mass in top $k$ tokens in this case, as shown in the last row of the figure. This observation aligns with our findings: in runs without decreasing entropy, we do not observe a significant decline in win probability.

**Learning curves with 35B model on the BINARIZEDPREF dataset.** To demonstrate the generalisability of our findings, we perform additional experiments using the 35B model on the BINARIZEDPREF dataset. The results align well with our previous observations. Figure 10 illustrates several key findings:

1. **Likelihood and Performance Correlation:** Similarly, results from larger model sizes suggest that higher likelihoods for better completions and larger gaps between better and worse completions do not necessarily lead to improved model performance, as shown in the first and second rows of the figure.

2. **Likelihood and Cross-Input Diversity:** Lower completion likelihood tends to enhance the models' Cross-Input Diversity. Specifically, the curve with a lower better completion likelihood generally tends to have a higher Cross-Input Diversity.

3. **Entropy and Over-Optimisation:** A decrease in entropy over the top-$k$ tokens (Per-Input Diversity) appears to indicate over-optimisation for diversity. For instance, the pink lines for DPO and IPO show a clear drop in entropy after 500 steps, accompanied by a decline in win probability.

4. **Probability Mass Distribution:** Similarly, we do not observe a decrease in probability mass in top $k$ tokens in this case, as shown in the last row of the figure.

**Training Negative Log-Likelihood Loss on better completions has limited influence on the model when it cannot affect completion likelihood.** To demonstrate the generalisability of our findings, we perform further experiments with 35B models on the BINARIZEDPREF dataset. As shown in Figure 11, we experiment with DPO using three different values of $\beta$, adding NLL loss as an auxiliary loss with four distinct coefficients for each $\beta$. Similarly to our findings in the main text, results indicate that when there is limited impact on the likelihood, the NLL loss has minimal impact on model performance. Training Negative Log-Likelihood Loss on better completions remains susceptible to over-optimisation.

Table 4: Examples for TRIVIAQA.

| |
|---|
| Question: {question} |
| Reference Answer: {reference_answer} |
| Model Output: {model_output} |
| |
| Evaluate the correctness of the model output compared to the reference answer. |
| Respond with EXACTLY ONE of the following options: |
| - Yes |
| - No |
| - Unsure |
| |
| Guidelines: |
| - Yes: If the model output is correct or equivalent to the reference answer. |
| - No: If the model output is incorrect or contradicts the reference answer. |
| - Unsure: If you can't determine the correctness or if there's insufficient information. |
| |
| Do not provide any explanation or additional text. Your entire response must be a single word. |
| |
| Your response: |

Discussion about Relationship Between KL and Completion likelihood. We report the $L_2$ loss between the policy model and the reference model with respect to the likelihood. This serves as a proxy for KL divergence, as both measure the divergence between the policy and reference models. While we could not generate a direct KL vs. Likelihood plot due to access restrictions, this proxy analysis allows us to provide relevant insights without requiring additional model retraining.

As shown in Figure 12, our experiments reveal that likelihood does not strictly correlate with the $L_2$ loss: lower likelihood (higher cross-entropy loss) does not necessarily correspond to a higher $L_2$ loss. This result suggests that the relationship between the likelihood of preferred completions and the divergence between the models is more nuanced than a simple monotonic association. In particular, the observed patterns reinforce the idea that likelihood and KL divergence, while connected under specific assumptions, are not directly interchangeable.

## C  FURTHER INVESTIGATIONS FOR QUESTION ANSWERING TASKS

**Case studies for NATURALQUESTIONSOPEN and TRIVIAQA tasks.** Table 5 provides two examples for NATURALQUESTIONSOPEN and TRIVIAQA tasks, respectively.

**LLM-as-a-Judge for the NATURALQUESTIONSOPEN task.** We implement a more flexible evaluation method to understand the potential issue of stylistic variations in answers. Instead of relying on exact string matching, which can be overly rigid, we employ an LLM-as-a-Judge using the

Table 5: Model output examples for NATURALQUESTIONSOPEN and TRIVIAQA.

| | Examples for NATURALQUESTIONSOPEN | |
|---|---|---|
| **Field** | **Content** | **$F_1$ Word** |
| Question | Where is dakar located on the world map? | – |
| High Likelihood Answer | Senegal | 100.0% |
| Mid Likelihood Answer | Dakar is the capital of Senegal and is located in West Africa. It is situated on the western coast of the country, on the Atlantic Ocean. | 8.7% |
| | Examples for TRIVIAQA | |
| **Field** | **Content** | **$F_1$ Word** |
| Question | How many Rings of Power were there, in total? | – |
| High Likelihood Answer | 20 | 100.0% |
| Mid Likelihood Answer | There were 20 Rings of Power in total, 3 of which were given to the Elves, 7 to the Dwarves, and 9 to the Men. | 8.7% |

GPT4o model. As shown in Table 4, this LLM-based evaluation system is presented with the original question, the reference answer, and the model's output. It then assesses whether the model's output is correct, incorrect, or if there's not enough information to make a determination, responding with "Yes", "No", or "Unsure" respectively. We compute the model performance based on the percentage of "Yes". Figure 13 shows the model performance on the ULTRAFEEDBACK dataset using the 7B model. Our analysis reveals that while the LLM-as-a-Judge evaluation method demonstrates a trend similar to the $F_1$ score, it consistently yields higher performance metrics.

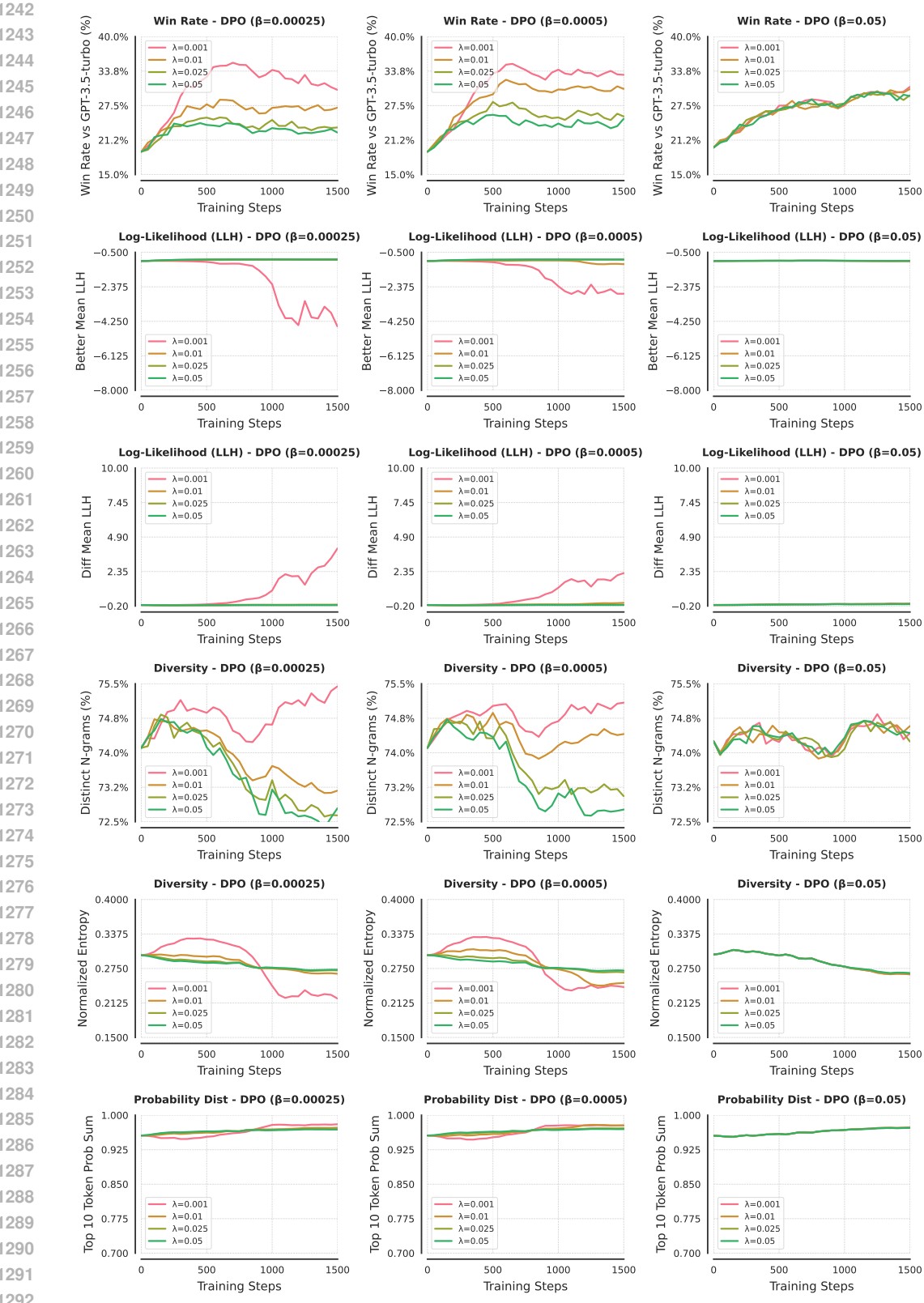

Figure 8: **Learning curves across training steps for various metrics**. Results are reported for the 7B models using IPO on the ULTRAFEEDBACK dataset with varying values of $\tau$ and $\lambda$.

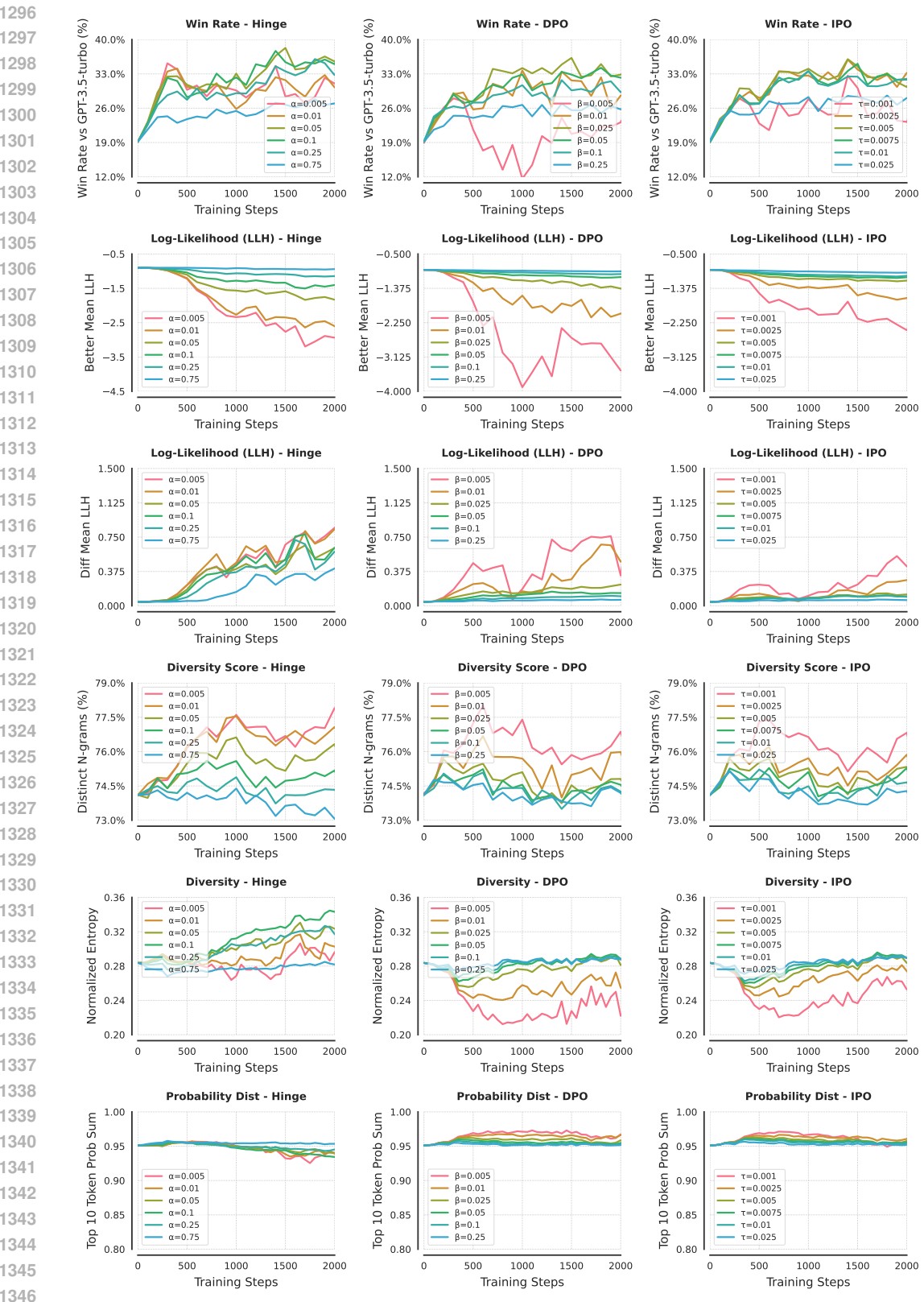

Figure 9: **Learning curves across training steps for various metrics**. Results are reported for the 7B models using the Hinge, DPO, and IPO on the BINARIZEDPREF dataset.

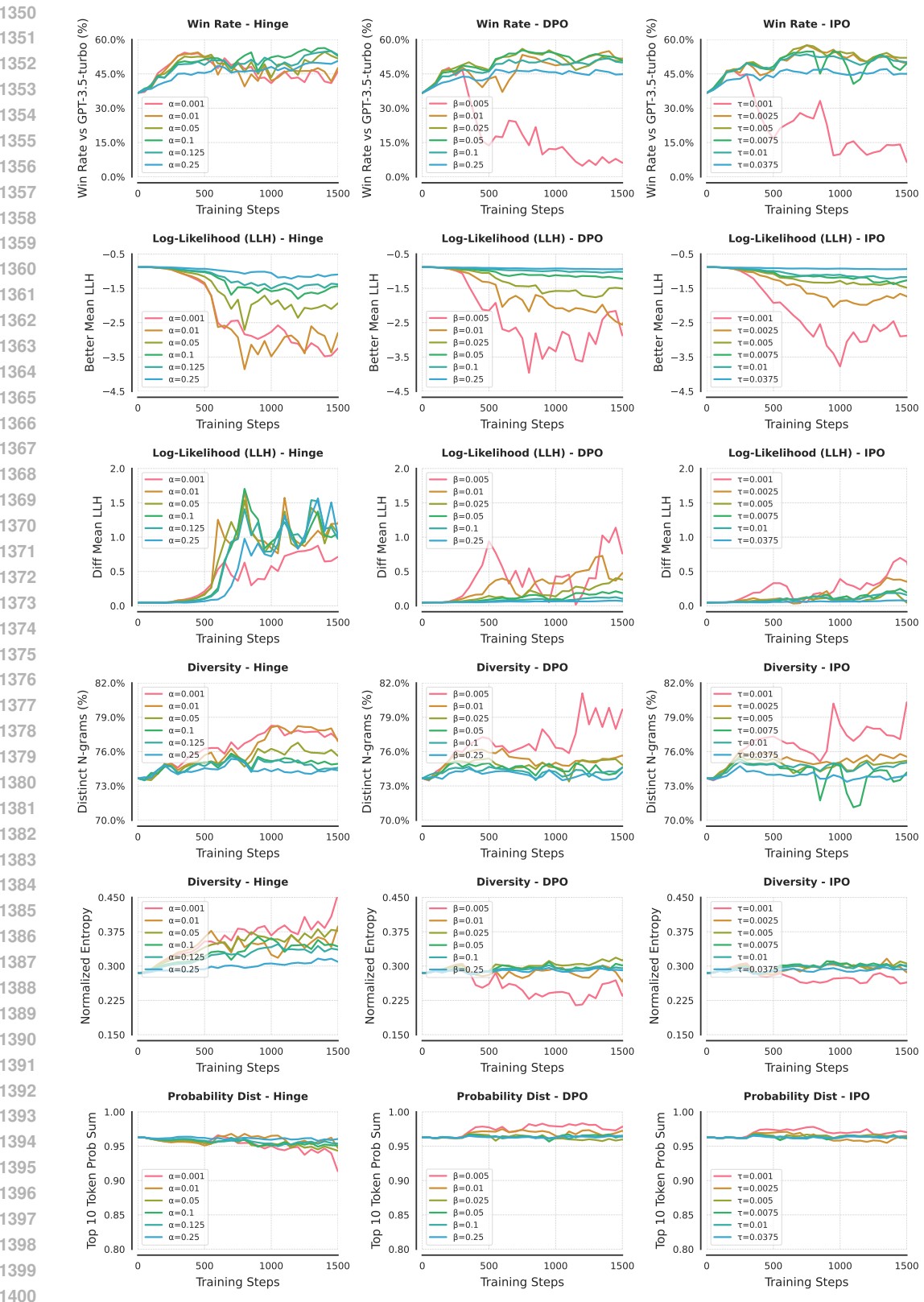

Figure 10: **Learning curves across training steps for various metrics**. Results are reported for the 35B models using the Hinge, DPO, and IPO on the BINARIZEDPREF dataset.

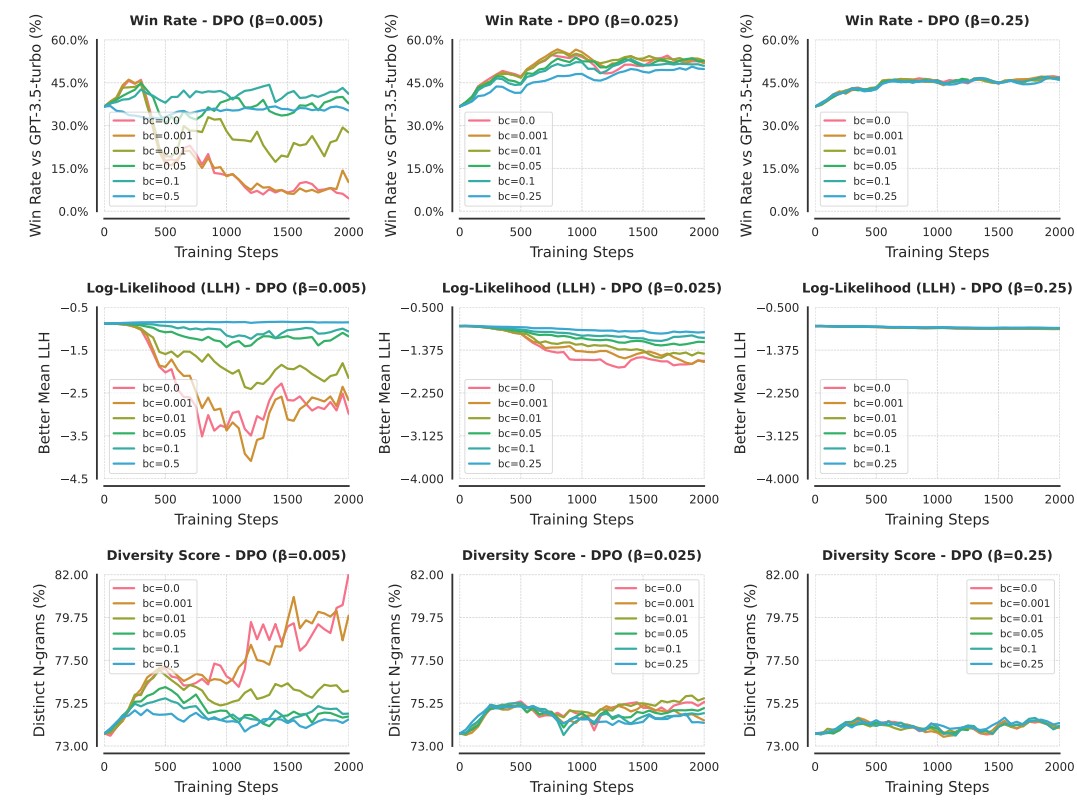

Figure 11: Control Likelihood via training on better completion on the BINARIZEDPREF dataset, using the 35B model. When different runs have similar likelihoods, the win probability and diversity of their model outputs tend to follow the same trend throughout training.

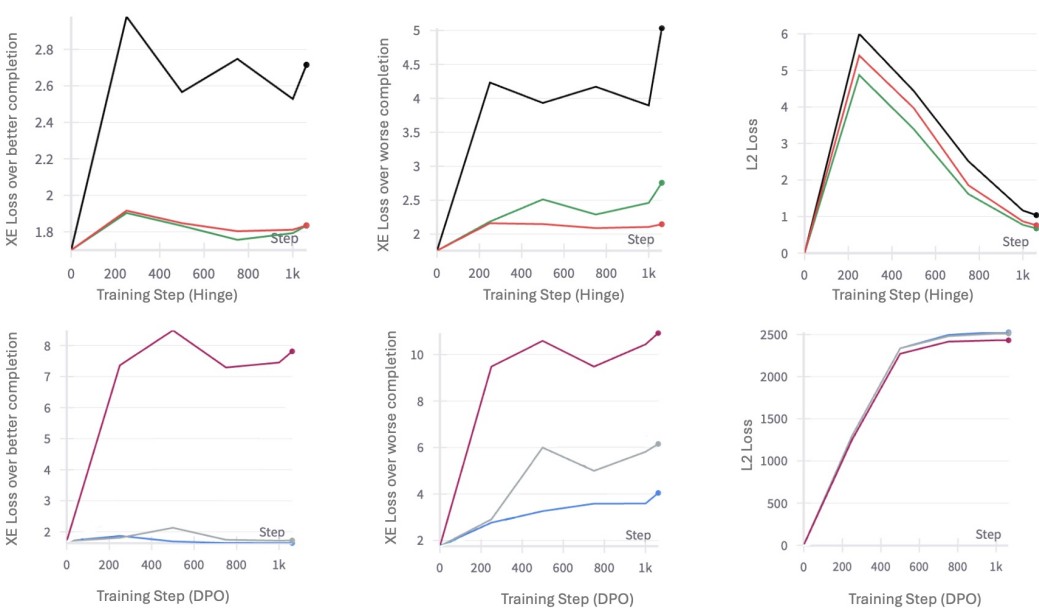

Figure 12: Our results indicate that completion likelihood does not strictly correlate with the $L_2$ loss: lower likelihood (higher cross-entropy loss) does not necessarily correspond to a higher $L_2$ loss.

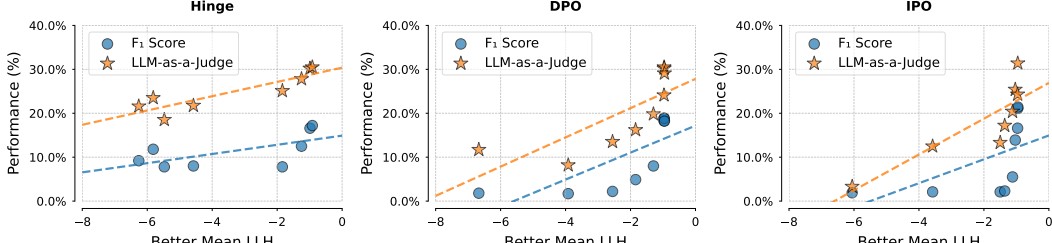

Figure 13: NATURALQUESTIONSOPEN vs Better Mean LLH on the ULTRAFEEDBACK dataset using the 7B model. The $F_1$ score and LLM-as-a-Judge results are reported.

