# OpenReview forum: "Understanding Likelihood Over-optimisation in Direct Alignment Algorithms"
_ICLR.cc/2025/Conference — Submitted to ICLR 2025_

### Official Review · Reviewer_tR7t · 2024-10-17

**Soundness:** 2
**Presentation:** 1
**Contribution:** 2
**Rating:** 3
**Confidence:** 4

**Summary:**

The authors explored the effect of increasing the probability of chosen sequences on the overoptimization of direct alignment algorithms (DAA). They concluded that an increased gap between chosen and rejected sequences could lead to overoptimization and linked overoptimization with reduced diversity of samples.

**Strengths:**

- The initial problem of overoptimization is important, and solving it is crucial for the field of alignment.
- Understanding signals of overoptimization is helpful and allows for faster evaluation. Instead of waiting for evaluation with LLMs, we can detect overoptimization through reduced entropy and decreasing mass in top-k tokens.

**Weaknesses:**

- While the authors provided implementation details, it is unclear why they used a 7B model with closed weights when a wide range of open-source models of this size is available.
- My main concern is the limited novelty of the obtained results. From what I see in Figure 2, while the gap between chosen and rejected sequences is increasing for small $\beta$, the likelihood of chosen sequences is decreasing. This aligns with the observations of [1], as probability mass moves towards out-of-distribution (OOD) examples (and to avoid overoptimization, we should prevent leakage of mass to OOD sequences). Therefore, the findings do not seem to present new insights. When demonstrating that increased probabilities for chosen sequences can also lead to overoptimization, it is important to explore the probabilities associated with OOD examples. From this perspective, the main novelty of the paper lies in understanding the mechanism of detecting overoptimization via reduced diversity and aligning these observations with the impact on performance across various tasks, but this aspect is poorly explored.
- The presentation of results is difficult to read. The captions of large figures (2, 3, 4) do not contain useful information for understanding, making some metrics hard to interpret. Additionally, the analysis of these results is far from the figures themselves.

[1] Scaling Laws for Reward Model Overoptimization in Direct Alignment Algorithms, Rafailov et. al.

**Questions:**

See Weaknesses

---

> ### Author Response · Authors · 2024-11-19
> **Author Rebuttal by Authors (1/2)**
>
> We appreciate the effort and time of the reviewer (tR7t). We would like to address the reviewer's valuable feedback as follows:
>
> > While the authors provided implementation details, it is unclear why they used a 7B model with closed weights when a wide range of open-source models of this size is available.
>
> We acknowledge that using open-source models would enhance reproducibility, but due to the constraints of this study, we focused on a controlled setup. In Section 4.1, we offer full details on the models used (7B and 35B parameter models) and the datasets (ULTRAFEEDBACK and BINARIZEDPREF), including exact versions and sizes. While the 7B model is closed-source due to the use of proprietary models or data, we have described these in as much detail as possible. Furthermore, we posit that our findings are likely generalisable to other LLMs, as most LLMs (e.g., Llama, Gemini) are based on standard transformer architectures (Vaswani et al., 2017). For example, the Llama model family has very standard features such as RoPE embeddings (Su et al., 2024).
>
> We also refer the reviewer to the response to reviewer dJek. In summary, the 35B model has openly available weights, and we considered it of scientific importance to compare the 35B model to a smaller model from the same family, rather than one where we have limited or no visibility into the full data selection and training process. There is considerable value in the ability to compare our observations across model sizes. There is also the (unfortunate but very real) more practical limitation of computing resources. The Llama3 models are available in 8B and 70.6B, and would not have allowed us to carry out the full suite of experiments planned. While we value and are aligned with the reviewer’s point of view, we consider an experimental setup with a closed 7B and an open-weight 35B model a fair compromise for the sake of sharing innovative research, and we hope that the reviewer can be convinced to reconsider accepting our research contributions in this spirit.
>
> > This aligns with the observations of [1], as probability mass moves towards out-of-distribution (OOD) examples (and to avoid overoptimization, we should prevent leakage of mass to OOD sequences). Therefore, the findings do not seem to present new insights.
>
> We respectfully disagree with this interpretation. The reviewer suggests that [1] posits probability mass moves toward out-of-distribution (OOD) examples and that preventing this leakage is necessary to avoid overoptimisation. In contrast, our work presents an opposing view: we demonstrate that a slight decrease in the likelihood of chosen sequences—effectively redistributing probability mass—can improve output diversity and ultimately enhance model performance. This perspective challenges the idea that avoiding OOD leakage is the sole factor in alignment. Instead, our findings highlight the critical role of balancing likelihood adjustments across chosen and OOD sequences, providing fresh insights into over-optimization mechanisms.
>
> >  When demonstrating that increased probabilities for chosen sequences can also lead to overoptimization, it is important to explore the probabilities associated with OOD examples.
>
> We argue that our findings serve to indirectly address the reviewer's concern. Since the total probability mass must sum to one, increasing the probabilities for chosen sequences necessarily reduces the probability of OOD examples. Thus, this exploration is implicitly addressed through the principle of probability mass conservation. Our findings emphasize that excessive likelihood increases for chosen sequences can still lead to over-optimisation, underscoring the importance of carefully managing likelihood shifts during alignment training. Nonetheless, we consider this a point of interest and would appreciate the opportunity to discuss this further with the reviewer.

---

> > ### Author Response · Authors · 2024-11-19
> > **Author Rebuttal by Authors (2/2)**
> >
> > > From this perspective, the main novelty of the paper lies in understanding the mechanism of detecting overoptimization via reduced diversity and aligning these observations with the impact on performance across various tasks, but this aspect is poorly explored.
> >
> > Detecting over-optimisation through reduced diversity is a key part of our contributions, and we strongly argue that this aspect has been extensively explored in our work. We have tested a variety of methods, datasets, and model sizes, and we tracked diversity across the entire training process. Furthermore, we provide two distinct diversity measures to ensure a thorough analysis. Given the comprehensiveness of our approach, we consider that this work provides both a deep and wide exploration of aligning diversity effects with downstream performance across a reasonable selection of models, direct alignment algorithms and evaluation tasks. We do however note that the clarity with which this is presented can be improved, and we will provide a revised version in the camera-ready version as we need some time to rethink how best to convey this message. We greatly appreciate the reviewer’s support in helping to frame the core contributions here.
> >
> > > The presentation of results is difficult to read. The captions of large figures (2, 3, 4) do not contain useful information for understanding, making some metrics hard to interpret. Additionally, the analysis of these results is far from the figures themselves.
> >
> > Thank you for this helpful feedback! We’ve updated the captions for Figures 2 and 4 to include more detailed explanations, making them clearer and easier to follow for readers. We hope these revisions enhance the overall understanding of our results.
> >
> > We thank the reviewer once again for their extremely helpful and constructive feedback. We believe that we’ve taken the key points on board and are working on addressing them in a revised version. We strongly agree with the reviewer that solving the problem of over-optimisation is crucial for the field of alignment and thank the reviewer for their contributions in strengthening this presentation of this work. We hope that this is sufficient to reconsider indicating an acceptance score following our revisions and look forward to further discussions.

---

> > > ### Comment · Reviewer_tR7t · 2024-11-22
> > > **Response on rebuttal**
> > >
> > > > In contrast, our work presents an opposing view: we demonstrate that a slight decrease in the likelihood of chosen sequences—effectively redistributing probability mass
> > >
> > > As I mentioned in my review, I am concerned about Figure 2. Do I understand correctly that this phenomenon can be observed around the 600th step, where the win-rate is at its peak while the probabilities of chosen sequences are slightly reduced?
> > >
> > > Also, from what I see in this figure, it does not seem that the entire observed results could be considered "an opposing view". Rather, we see that during optimization, it could be beneficial to slightly reduce the probabilities of chosen sequences. However, they further decrease over the course of training. If so, do you have any explanations for this behavior? As a researcher in the alignment field, I am more interested in providing insights into the behavior of methods rather than results merely indicating that some behavior could occur.
> > >
> > > > Given the comprehensiveness of our approach, we consider that this work provides both a deep and wide exploration of aligning diversity effects with downstream performance across a reasonable selection of models, direct alignment algorithms and evaluation tasks
> > >
> > > When discussing different tasks, it is essential to consider that they are not equal. The Ultrafeedback dataset consists of rejected sequences that are slightly worse than the chosen ones. In contrast, there are datasets that contain rejects that are much worse than the chosen ones, and the behavior of the training dynamics would differ (e.g., HH). From this perspective, I am not fully satisfied with the insights obtained from this work, as it lacks depth.
> > >
> > > > but due to the constraints of this study, we focused on a controlled setup.
> > >
> > > The authors' response does not cover my concerns. I do not understand what constraints prevented the authors from using widely accessible models for evaluation. This is also highlighted by reviewer "dJek". Also, while writing this response, I noticed that the BINARIZEDPREF dataset is also proprietary data. While the authors provided details of this dataset in the rebuttal revision, I believe that the use of such datasets greatly reduces the reproducibility of the work and limits its contribution. For example, as noted above, not all datasets are equal, and I simply do not understand what the chosen and rejected sequences look like. This limits insights into "how likelihoods on this data behave".

---

> ### Author Response · Authors · 2024-11-23
> **Author Rebuttal by Authors**
>
> We appreciate your thoughtful comments and the opportunity to clarify and expand on the points raised. Below, we address your concerns in detail.
>
> > As I mentioned in my review, I am concerned about Figure 2. Do I understand correctly that this phenomenon can be observed around the 600th step, where the win-rate is at its peak while the probabilities of chosen sequences are slightly reduced?
>
> Thank you so much for your question. Yes, that is correct.
>
> > Also, from what I see in this figure, it does not seem that the entire observed results could be considered "an opposing view". Rather, we see that during optimization, it could be beneficial to slightly reduce the probabilities of chosen sequences. However, they further decrease over the course of training. If so, do you have any explanations for this behavior? As a researcher in the alignment field, I am more interested in providing insights into the behavior of methods rather than results merely indicating that some behavior could occur.
>
> Thank you so much for your question.
>
> **Why does the reduced likelihood improve model performance?** A reduced completion likelihood gives the model a better chance to select alternative tokens. As shown in Figure 2, the probabilities of the chosen sequences align with the model’s Cross-Input Diversity, which enhances its ability to generalise to unseen scenarios. This is reflected in an increased win probability, indicating improved performance.
>
> **Why does overly reducing the likelihood harm model performance?**  At a certain point in training, reducing the completion likelihood too much can negatively impact the model's performance. For example, as explained in Section 4.4, during the early stages of training, Per-Input Diversity (measured by entropy over the top-k tokens, where k = 10) increases. This indicates a broader distribution of selected tokens and a more uniform output distribution for the next token prediction. Considering that the better completion likelihood is decreasing across the training, the increase of entropy at the beginning phase indicates that those tokens from better completion have a higher probability at the initial policy model over other tokens in the top k. The decrease better completion likelihood gives the model a better chance to select other tokens, which increases diversity and enhances generalisation, as reflected in the win probability. However, this trend reverses after a certain point. As Per-Input Diversity (entropy) starts decreasing, the model begins to over-prioritise certain tokens. This suggests that those tokens in the better completion now have an overly low likelihood, lower than other tokens in the top k. Despite this, Cross-Input Diversity keeps increasing, which indicates that the model is still generating diverse outputs, but now it includes tokens that are less relevant or nonsensical, i.e., tokens that humans do not prefer.
>
> > When discussing different tasks, it is essential to consider that they are not equal. The Ultrafeedback dataset consists of rejected sequences that are slightly worse than the chosen ones.  (...) I do not understand what constraints prevented the authors from using widely accessible models for evaluation.
>
> We appreciate your concern regarding the use of proprietary datasets and their potential impact on reproducibility. We want to clarify that significant portions of our work are fully reproducible:
> - Our study includes the open-sourced preference learning dataset UltraFeedback, which has been validated for its quality and utility, as evidenced by its successful use in training the open-source model Zephyr.
> - The 35B model weights are openly available. Our methodology is detailed enough to enable replication with similar open models (e.g., models like Aya-Expanse-8B that share our architectural foundations)
>
> The decision to use the BINARIZEDPREF dataset, while proprietary, was motivated by its unique characteristics that complement the publicly available dataset, allowing us to explore a broader range of alignment datasets. Additionally, we provided extensive details on this dataset in the rebuttal revision, including its construction and the quality differences between chosen and rejected sequences. We believe this transparency mitigates some of the limitations associated with proprietary datasets.
>
> It is also worth noting that the use of proprietary resources is common in the research community. For example, numerous published papers from Google rely on proprietary datasets and models, such as PaLM. We believe this aligns with standard guidelines for evaluating work based on its scientific contributions rather than the openness of every resource, particularly when sufficient details are provided to ensure reproducibility and understanding.
>
> If anything is unclear or requires further elaboration, please don't hesitate to ask, and we would be more than happy to discuss and refine our approach.

---

> > ### Comment · Reviewer_tR7t · 2024-11-24
> > **Response**
> >
> > > A reduced completion likelihood gives the model a better chance to select alternative tokens. As shown in Figure 2, the probabilities of the chosen sequences align with the model’s Cross-Input Diversity, which enhances its ability to generalise to unseen scenarios. This is reflected in an increased win probability, indicating improved performance.
> >
> >
> > As mentioned in other reviews, my main concern regarding the claim that this process offers an orthogonal view on overoptimization is that, with the reduction of NLL (which is clearly observable), there will also be a clearly observable increase in KL divergence. From this perspective, Figure 1 could be interpreted as a typical scaling law of overoptimization (although flipped across the vertical axis). Even if we assume that the L2 norm from the added plot is an approximation for KL divergence (which is actually not the case; otherwise, natural gradients would not be necessary), this pattern could be observed, despite the figure's caption stating that it could not. Therefore, I remain unconvinced by the results presented in the paper. Additionally, I believe that these results should be plotted similarly to Figure 1, with KL divergence versus NLL(y_w).
> >
> > > It is also worth noting that the use of proprietary resources is common in the research community. For example, numerous published papers from Google rely on proprietary datasets and models, such as PaLM. We believe this aligns with standard guidelines for evaluating work based on its scientific contributions rather than the openness of every resource, particularly when sufficient details are provided to ensure reproducibility and understanding.
> >
> > I would like to highlight that the use of proprietary resources is *not* common in the research community but is rather common among corporations publishing their results. At this point, most of the results are not reproducible since half of the experiments were conducted with proprietary models and datasets. The lack of released source code makes validation of these results difficult.
> >
> > When discussing additional experiments, you explicitly state, "While we could not generate a direct KL vs. NLL(y_w) plot due to access restrictions..." Even if the authors were unable to perform additional experiments, how is the research community expected to replicate or extend these findings?
> >
> > > Additionally, we provided extensive details on this dataset in the rebuttal revision, including its construction and the quality differences between chosen and rejected sequences. We believe this transparency mitigates some of the limitations associated with proprietary datasets.
> >
> > A one-paragraph explanation of the dataset is *not* extensive detail. Compare it to the details provided with the ultrafeedback dataset (https://arxiv.org/pdf/2310.01377). Furthermore, I suggest that conducting experiments with a proprietary dataset, when a wide range of openly accessible datasets is available, is not ideal and could have been easily avoided.

---

> > > ### Author Response · Authors · 2024-11-25
> > > **Author Rebuttal by Authors**
> > >
> > > We thank the reviewer for engaging in valuable and insightful discussions, particularly over the weekend. As mentioned in the response to reviewer, dJek, we will commit to including the KL vs NLL scatterplot in the camera-ready version of the paper. We refer the reviewer to that response for additional detail.
> > >
> > > Regarding the points around openness and reproducibility, we feel that the reviewer is taking a stand on the principle of open research. We agree that reproducibility is a cornerstone of rigorous scientific research, and we appreciate the reviewer’s commitment to this principle. We are a group of researchers and scientists who have collectively made multiple open-source contributions, and remain committed to sharing our work openly and transparently. However, there are constraints beyond our control within which we need to work to ensure that the insights we learn can be shared back to the community. We would ask the reviewer to consider that taking the strict position that work that does not use solely open-source datasets or model weights should be rejected creates the opposite outcome that the community desires. Instead of moving towards increased openness and sharing of research, we move towards a scenario where industry labs have no avenue for sharing their research contributions. We believe that incentivising all researchers, including those working with operational constraints, to share their insights is the best path forward for the community at large, while continuing to incentivise the open release of resources wherever possible.
> > >
> > > To address each of the reviewer’s points directly:
> > > - **Proprietary dataset**. Our work primarily concerns a binarised version of the UltraFeedback dataset. This is a public dataset, sampled from 6 existing public datasets, with completions sampled from a range of commercial, open-weight, and open-source models. We validate our results and insights on the proprietary BinarizedPref dataset, and we share previously unreleased details of its construction. This validation is not central to our contributions, however, there is value in demonstrating that our findings also hold validity for datasets where there are no concerns of effects such leakage into pretraining. We could have opted to redact this from our contribution, but feel that this would be contrary to our shared desire for openness. We welcome any suggestions from the reviewer about any additional detail regarding this dataset that they feel would considerably strengthen our contributions.
> > > - **Proprietary models**. We use a 7B and 35B model in this work. The weights of the 35B model are openly available. We have already provided details of the weights of a 7B model that is architecturally identical to the model we investigated. When it came to model selection, we had specific criteria: models from the same family with similar pre- and post-training, model sizes in the ranges close to 7B and 35B to meet our compute budget, competitive models to ensure that any insights we gleaned were valuable moving forward, and openness. We are not aware of any other models that fit these requirements and kindly ask the reviewer to provide specifics or suggestions otherwise.
> > > - **Lack of released source code**. Our contributions are not code contributions. There are many open codebases in which our findings can be reproduced, and we provide full details for doing so. If there are any specific training details the reviewer considers missing, we would kindly ask that the reviewer highlights these such that we can add the detail in the relevant appendices.
> > >
> > > Finally regarding the reviewer’s comment that “`Even if the authors were unable to perform additional experiments, how is the research community expected to replicate or extend these findings?`”, we will note that our response clearly provides our reason being access restrictions. We believe that with the detail provided and sufficient computing access, researchers in the open community are more than able to replicate and build on our findings.
> > >
> > > We believe this work provides significant insights and advances that will benefit the community, and we hope the reviewers can consider the contributions in their entirety.

---

### Official Review · Reviewer_PsR8 · 2024-10-29

**Soundness:** 3
**Presentation:** 2
**Contribution:** 3
**Rating:** 8
**Confidence:** 3

**Summary:**

This paper provides a experimental analysis of the behaviors of margin-based alignment approaches, especially, they focus on the so-call over-optimisation issue. They claim that they are the first to explore the relationship between completion likelihood and performance in alignment algorithms. Their empirical finding is that, likelihood does not have a striong positive correlation with model performance, and it might be affected by diversity. Besides, they identify two indicators of overly generatinve diverse outputs.

**Strengths:**

## Originality
- Though the claim may not be novel, the systematic experiments are original and interesting.
- The reviewer appreciates the statistical metrics.

## Clarity
- This paper is well-written and densely organised.
- The figures are clearly plotted.

## Significance
- This paper may be able to clarify a controversial point, whether DAA should directly increase(decrease) the likelihood of accepted(rejected) completion? which the reviewer believes is a important question.

**Weaknesses:**

## Major
- The two indicators are so trivial, that people already know this before. Thus these two findings might not be counted as contributions.
- The experiments are restricted. Some DAAs with alternate objective (CPO, SimPO, ...) (such as length normalization) are not tested, making the claim less solid.

## Minor
- The curves are a bit confusing, especially Figure 4, making it hard to come to the authors' conclusion.

**Questions:**

- In Figure 3, why the DPO/IPO 7B ultrafeedback model cannot gain much improvement compared with initial checkpoint?

---

> ### Author Response · Authors · 2024-11-19
> **Author Rebuttal by Authors**
>
> We appreciate the effort and time of the reviewer (PsR8). We would like to address the reviewer's valuable feedback as follows:
>
> > The two indicators are so trivial, that people already know this before. Thus these two findings might not be counted as contributions.
>
> The reviewer raises an important point which we have clarified in the main text of the paper. We respectfully disagree with the observation that the indicators introduced are trivial. While intuitively plausible, these indicators have not been systematically formalised or studied in the context of DAAs prior to our work. Our contributions lie in:
> - Formalizing these indicators: Connecting diversity reductions and performance trends explicitly to over-optimization in DAAs.
> - Quantitative empirical validation: Demonstrating these trends across diverse datasets, models, and methods.
> If the reviewer is aware of prior studies explicitly formalising these indicators, we kindly request references for the appropriate acknowledgement. Even in the event that there exists previous work that we are unaware of that has studied these effects, we consider both the breadth and depth of experiments across which we validate these findings to be considerable strengths and a contribution in their own right.
>
> > The experiments are restricted. Some DAAs with alternate objective (CPO, SimPO, ...) (such as length normalization) are not tested, making the claim less solid.
>
> We respectfully note that our experiments include three popular DAAs: DPO, IPO, and SILC, which were selected based on prior work [1]. We will explore additional objectives like CPO and future work. We also refer the reviewer to the response to reviewer dJek where, in summary, we highlight the high compute resources to run such experiments and which do not directly strengthen the core findings of our work. We agree with the reviewer that extending our exploration to additional DAAs would be of significant interest, which further validates the contributions made in this work, but doing so is computationally infeasible and beyond the scope of this work. We thus leave such exploration to future work.
>
> [1] Scaling laws for reward model overoptimization in direct alignment algorithms.
>
> > In Figure 3, why the DPO/IPO 7B ultrafeedback model cannot gain much improvement compared with initial checkpoint?
>
> Our goal is not to train the model to optimize performance but rather to understand the relationship between likelihood and model performance. The limited improvement observed in the DPO/IPO 7B ultrafeedback model reflects our focus on studying the dynamics of over-optimisation, rather than fine-tuning for maximum performance gains.
>
> We thank the reviewer once again for their valuable and constructive review. We hope that our revised upload has served to address the main points for improvement identified and look forward to continuing to strengthen the clarity of the work through further discussion.

---

> > ### Comment · Reviewer_PsR8 · 2024-11-19
> > **Raise my rating to 8**
> >
> > Hi, thank you for your clarifications. I agree with that the two indicators are intuitively plausible while they have not been systematically formalised in the context of DAA. And I understand the high computation demand. Now my concerns resolved.
> >
> > I will raise my rating to $8$.

---

> > > ### Author Response · Authors · 2024-11-20
> > > **Thank you!**
> > >
> > > Thank you for your thoughtful feedback and raised rating. We’re glad our clarifications addressed your concerns.

---

### Official Review · Reviewer_dJek · 2024-11-03

**Soundness:** 2
**Presentation:** 3
**Contribution:** 2
**Rating:** 3
**Confidence:** 3

**Summary:**

This paper presents a detailed analysis of contemporary offline alignment methods, focusing on DPO, IPO, and Hinge. A comprehensive set of metrics, gathered during the alignment process for both a proprietary 7B model and the Cohere Command R 35B model, is employed to investigate the issue of over-optimization. Furthermore, the paper proposes using the entropy of the top-k tokens during generation for DPO and IPO, as well as the aggregate mass of the top-k tokens for Hinge, as indicators of over-optimization.

**Strengths:**

- Evaluation details are comprehensive.
- This work is useful for the community as it could be used to determine offline metrics to identify the best model at various steps.

**Weaknesses:**

- Over-optimization itself is already well-studied in [1]. Some results, like Figure 1, seem to be consequences of the high KL divergence. I think NLL for y_w and KL are highly correlated metrics. It would be useful to see a figure with KL on the x-axis and NLL(y_w) on the y-axis.
- While Figure 1 supports the claim that there is no correlation between NLL(y_w) and win rate in general, Figure 2 shows that, for the given method and hyperparameters, the best step can be determined by observing NLL(y_w). More precisely, I would suggest tracking the difference between NLL from the previous and current checkpoints. If this value becomes larger than a certain threshold, one can stop training and thereby obtain the strongest checkpoint. Therefore, from the reported I would say that tracking NLL(y_w) could be sufficient to detect over-optimization across all methods (instead of proposed methodology in Section 4.4, which depends on method).
- The paper presents an extensive amount of metrics during the training procedure; however, there is no clear criterion to detect over-optimization. First of all, the proposed methods in Section 4.4 are highly dependent on alignment methods, and if one were to use other popular methods (like KTO, ORPO, etc.), it is not clear which metric would serve as a flag for over-optimization. Additionally, it is unclear if these results remain the same for different models (see next weakness point).
- The choice of models is in question. Results obtained from a closed-source 7B model may not generalize to widely used open-weight models with similar sizes like LLAMA, Gemma, or Mistral. Additionally, this has a negative impact on the reproducibility of the experiments.

[1] Scaling Laws for Reward Model Over-optimization in Direct Alignment Algorithms (Rafailov et al.)

**Questions:**

Lower entropy usually indicates that a model is overfitted. Could this fact indicate that the UltraFeedback dataset is not sufficiently diverse, and therefore the model could memorize patterns that are not useful in general? Does the issue of over-optimization hold across different datasets?

---

> ### Author Response · Authors · 2024-11-19
> **Author Rebuttal by Authors (1/2)**
>
> We appreciate the effort and time taken by the reviewer (dJek) to provide such a detailed and constructive review. We respond to the main weaknesses and questions below and incorporate much of this in a revised version of the paper, which we believe considerably strengthens our contribution.
>
> > Over-optimization itself is already well-studied in [1]. Some results, like Figure 1, seem to be consequences of the high KL divergence. I think NLL for y_w and KL are highly correlated metrics. It would be useful to see a figure with KL on the x-axis and NLL(y_w) on the y-axis.
>
> Thank you for the suggestion. While prior work [1] has examined over-optimisation through the lens of KL divergence, our analysis uncovers a more intricate relationship. Specifically:
> - **Rafailov et al.** focuses on KL divergence as a primary metric for assessing over-optimization. In contrast, our work introduces a multi-dimensional approach, analysing patterns in likelihood, entropy dynamics, and probability mass distribution.
> - **KL divergence vs. completion likelihood**: KL divergence does not directly correlate with the completion likelihood. Both increases and decreases in completion likelihood can result in higher KL divergence from the reference model. KL divergence is more about how far the model should move, while our likelihood analysis is more about which direction the model should move.
>
> We will include an additional figure plotting KL versus NLL(y_w) in future work, which we believe will further clarify this relationship and strengthen our conclusions.
>
> > While Figure 1 supports the claim that there is no correlation between NLL(y_w) and win rate in general, Figure 2 shows that, for the given method and hyperparameters, the best step can be determined by observing NLL(y_w). More precisely, I would suggest tracking the difference between NLL from the previous and current checkpoints. If this value becomes larger than a certain threshold, one can stop training and thereby obtain the strongest checkpoint. Therefore, from the reported I would say that tracking NLL(y_w) could be sufficient to detect over-optimization across all methods (instead of proposed methodology in Section 4.4, which depends on method).
>
> We respectfully disagree with the suggestion that tracking NLL(y_w) differences could serve as a universal stopping criterion. Our experiments show that:
> - Optimal NLL ranges differ significantly across DAAs, making a universal threshold impractical (evident in Figures 2-4).
> - Using NLL thresholds would require method-specific calibration, defeating the purpose of having a general detection mechanism.
> - Consider the comparison between win rates in the first row of Figure 2 and NLL(y_w) in the second row of Figure 2, across all three DAAs, we can consistently see that win rates peak earlier than tracking NLL(y_w) would suggest, even accounting for effects like smoothing/moving averages or deltas to previous checkpoints. NLL(y_w) also fluctuates considerably, and would require a separate representative held-out validation set, and method-specific thresholds as mentioned above. It can and should be used as a signal, but relying on it alone is suboptimal.
>
> In contrast, our proposed method provides an algorithm-agnostic framework by leveraging additional signals such as diversity and provides signals that correlate more reliably with likelihood over-optimisation across different DAAs, which is also particularly beneficial in the common situation of selecting between different DAAs for best performance.

---

> > ### Author Response · Authors · 2024-11-19
> > **Author Rebuttal by Authors (2/2)**
> >
> > > The paper presents an extensive amount of metrics during the training procedure; however, there is no clear criterion to detect over-optimization. First of all, the proposed methods in Section 4.4 are highly dependent on alignment methods, and if one were to use other popular methods (like KTO, ORPO, etc.), it is not clear which metric would serve as a flag for over-optimization. Additionally, it is unclear if these results remain the same for different models (see next weakness point).
> >
> > Our experiments include three widely-used DAAs: DPO, IPO, and SLiC, which were selected based on prior work [1]. Regarding ORPO and KTO:
> > - ORPO can be viewed as a combination of DPO and SFT over preferred completions, which our experiments already effectively cover.
> > - While KTO would be interesting to study, our current results with major DAA families suggest our findings would likely extend to it as well. While we could extend this work to other popular methods, each set of experiments uses a considerable amount of compute resources and does not serve to substantially strengthen the main contributions of our work. We therefore leave testing the generalisability of our findings to other methods for future work.
> > - The consistency of our findings across multiple methods provides strong evidence for their generalizability.
> >
> > We nonetheless think that the reviewer raises an important point here, and we address this in an updated version of the discussion section of our paper.
> >
> > [1] Scaling laws for reward model overoptimisation in direct alignment algorithms.
> >
> >
> > > The choice of models is in question. Results obtained from a closed-source 7B model may not generalize to widely used open-weight models with similar sizes like LLAMA, Gemma, or Mistral. Additionally, this has a negative impact on the reproducibility of the experiments.
> >
> > While we acknowledge the reviewer's concern about using closed-source models, we believe our experimental setup and findings remain valuable for several reasons:
> > - The underlying architecture of our models follows standard transformer designs (Vaswani et al., 2017) similar to those used in popular open-source models. Key components like RoPE embeddings (Su et al., 2024) are shared across many modern LLMs including Llama and others
> > - Our detailed documentation of model specifications (Section 4.1) and experimental setup enables understanding and adaptation of our findings.
> > - The phenomena we observe are likely architecture-agnostic, as they stem from fundamental aspects of alignment optimisation rather than model-specific features.
> > - There is also the (unfortunate but very real) more practical limitation of computing resources. The Llama3 models are available in 8B and 70.6B, and would not have allowed us to carry out the full suite of experiments planned. While we value and are aligned with the reviewer’s point of view, we consider an experimental setup with a closed 7B and an open-weight 35B model a fair compromise for the sake of sharing innovative research, and we hope that the reviewer can be convinced to reconsider accepting our research contributions in this spirit.

---

> > > ### Comment · Reviewer_dJek · 2024-11-22
> > >
> > > Thank you for the answers, which partially address my concerns. However, I am still not convinced about some points.
> > >
> > > **KL vs NLL(y_w)**
> > > > KL divergence does not directly correlate with the completion likelihood.
> > >
> > >
> > > I agree that there is no strong rule for this; however, I believe that it is the most intuitive result you could get. On one hand, all DAAs methods increase the likelihood of the chosen samples during training. On the other hand, the more steps you take towards minimizing the non-KL term, on average, the more KL you get (taking aside LR and other hyperparameters).
> > > > We will include an additional figure plotting KL versus NLL(y_w) in future work, which we believe will further clarify this relationship and strengthen our conclusions.
> > >
> > >
> > > I believe this figure is needed in your current work to support your claims (as you mentioned above).
> > > > **Rafailov et al.** focuses on KL divergence ...
> > >
> > >
> > > Again, showing no correlation between NLL(y_w) and KL will clearly demonstrate the difference between your work and the work by Rafailov et al.
> > >
> > > **Signals of overoptimization**
> > > > Optimal NLL ranges differ significantly across DAAs, making a universal threshold impractical (evident in Figures 2-4).
> > >
> > >
> > > The bend in the NLL shown in Figures 2-4 is a promising indicator. Additionally, it is important to mention that the signal metrics used by the authors vary among the different DAAs.
> > >
> > > **DAAs selection**
> > > > ORPO can be viewed as a combination of DPO and SFT over preferred completions, which our experiments already effectively cover.
> > >
> > >
> > > I belive that non-likelihood term of ORPO
> > >
> > >  $L_{OR}$=$-\log \sigma(\log \frac{{\boldsymbol{odds}}(y_w \mid x)}{\boldsymbol{\boldsymbol{odds}}(y_l \mid x)})$
> > >
> > > can not be viewed as a DPO loss. At least because there is no $\pi_{ref}$ involved, in contrast to DPO loss. Can the authors clarify on that?
> > >
> > > **Model choice**
> > >
> > > While all architectures of modern LLMs are approximately the same, the results of your work will mainly be used in open-source LLMs alignment. Closed-source models harm reproducibility and introduce unnecessary bias in metrics, compared to open-source LLMs.

---

> > > > ### Comment · Reviewer_tR7t · 2024-11-22
> > > > **Comment**
> > > >
> > > > > We will include an additional figure plotting KL versus NLL(y_w) in future work, which we believe will further clarify this relationship and strengthen our conclusions.
> > > >
> > > >
> > > > It is evident that NLL(y_w) can be seen as an approximation of the KL divergence. From this perspective, the results in Figure 1 show exactly the same overparameterization laws as those presented in [1]. I am curious why these results are being deferred to future work, as they could directly demonstrate a potential lack of novelty.

---

> > > > ### Author Response · Authors · 2024-11-24
> > > > **Author Rebuttal by Authors (1/2)**
> > > >
> > > > We greatly appreciate your detailed feedback and thoughtful suggestions. Below, we address your concerns point by point.
> > > >
> > > > > KL vs NLL(y_w)
> > > >
> > > > We thank the reviewer for highlighting this important relationship. In response, we have added a figure plotting to the paper to directly support our claims.
> > > >
> > > > **Discussion about Relationship Between KL and NLL(y_w)**. We appreciate the reviewer’s comments on the relationship between KL divergence and NLL(yw)\text{NLL}(y_w)NLL(yw​).
> > > > - *The approximation between average log-likelihood of preferred responses and forward KL only hold naturally under the strong assumption that the reference policy reflects the preferred response distribution (e.g. π_ref is the SFT model trained on preferred completions).* Rafailov's work [1] assumes that the model is instruction-tuned on preferred completion before the DAA, and that the likelihood of preferred completions only decreases via the training, in which case NLL is approximate for KL. However, previous works have shown that likelihood of preferred completions could also be increased (for example, see Figure 2 and 3 in [2]).
> > > > - *The likelihood of preferred completions is NOT simply another form of KL divergence. As long as we train the policy model based on the reference model, the KL divergence will increase, regardless of whether we increase or decrease the likelihood of a better completion.* Rafailov's work addresses how far (in terms of KL) we should move the model away from the reference model. On the other hand, our paper focuses on how we can use this KL budget to adjust the likelihood.
> > > >
> > > > **Additional Plots.** We apologize that we were not clearer about the KL vs. NLL(y_w) relationship in our initial response. Unfortunately, we no longer have access to the computational resources required to generate this plot within the rebuttal period, as it would involve retraining models with additional tracking metrics. For this reason, we suggested including this analysis as future work—not to evade the question, but because we believe it merits a thorough and focused investigation.
> > > >
> > > > In response to the reviewer’s comments, we report the $L_2$ loss between the policy model and the reference model with respect to the likelihood. This serves as a proxy for KL divergence, as both measure the divergence between the policy and reference models. While we could not generate a direct KL vs. NLL(y_w) plot due to access restrictions, this proxy analysis allows us to provide relevant insights without requiring additional model retraining.
> > > >
> > > > As shown in the attached figure, our experiments reveal that likelihood does not strictly correlate with the $L_2$​ loss: lower likelihood (higher cross-entropy loss) does not necessarily correspond to a higher $L_2$ loss. This result suggests that the relationship between the likelihood of preferred completions and the divergence between the models is more nuanced than a simple monotonic association. In particular, the observed patterns reinforce the idea that likelihood and KL divergence, while connected under specific assumptions, are not directly interchangeable.
> > > >
> > > > We have added this detail to a revised version of our paper.
> > > >
> > > > **Novel Contributions Beyond Prior Work.** We thank the reviewer for highlighting the importance of situating our work within the context of Rafailov et al. We view their findings as foundational and explicitly build upon them in several key ways. Rafailov et al. explore how far (in terms of KL) the policy model should move from the reference model, framing KL divergence as a constraint on optimisation. Our work complements this by investigating how the KL budget can be used to adjust the likelihood of preferred responses while maintaining alignment with human preferences. This provides a different perspective on the trade-offs inherent in policy updates.
> > > >
> > > > Additionally, while Rafailov et al. focus on smaller-scale models, we extend their analysis to larger models, demonstrating that the patterns they identified persist at scale. Beyond scaling, our work contributes new insights, particularly through a detailed investigation of better and worse completion likelihoods—an aspect not addressed in prior work. This allows us to identify novel signals that can guide alignment and optimisation, offering a more granular understanding of the relationship between KL divergence and likelihood dynamics.
> > > >
> > > > ## References:
> > > > [1] Scaling Laws for Reward Model Overoptimization in Direct Alignment Algorithms.
> > > >
> > > > [2] Iterative Reasoning Preference Optimization. NeurIPS 2024.

---

> ### Author Response · Authors · 2024-11-24
> **Author Rebuttal by Authors (2/2)**
>
> > ORPO loss
>
>
> We acknowledge the reviewer’s observation regarding the distinction between ORPO’s loss formulation and DPO. In general, KL loss (where $\pi_{\text{ref}}$ is used) can be interpreted as a form of regularisation. In ORPO, KL regularisation is replaced with SFT (using a cross-entropy loss) over preferred completions. This forms the basis for our claim that ORPO can be viewed as a combination of DPO and SFT over preferred completions. This approach is conceptually similar to SLiC-HF [3], which also uses cross-entropy regularisation instead of KL.
>
> We hope this clarification helps to resolve any confusion, and we welcome further discussion on this point.
>
> > The bend in the NLL shown in Figures 2-4 is a promising indicator.
>
> We sincerely appreciate your suggestion that the bend in NLL shown in Figures 2-4 is a promising indicator. We have incorporated your insightful observation into our discussion, as it aligns closely with our core findings and adds further depth to the narrative of our paper. Additionally, we emphasise that identifying and leveraging such signals, including NLL, represents a novel aspect of our work that has not been explored in prior studies.
>
> > Model choice
>
> We thank the reviewers for their feedback. Below, we address the points raised to clarify our methodology and approach:
>
> **Reproducibility of Our Work**. A significant portion of our study is reproducible and utilizes open-source resources. Specifically:
>    - We employed the **UltraFeedback** dataset, an open-sourced preference learning dataset that has been validated in prior work for its quality and practical utility. This dataset successfully supported the training of Zephyr, a fully open-source model, underscoring its efficacy.
>    - The **35B model weights** used in our study are openly available, ensuring researchers can directly reproduce our results. Furthermore, our methodology is sufficiently detailed to enable replication with alternative open models such as **Aya-Expanse-8B**, which shares architectural foundations with our work.
>
> It is also worth noting that the use of proprietary resources is common practice in the research community. For example, numerous published papers from Google rely on proprietary datasets and models, such as PaLM, to advance research in large-scale language modelling. We believe this aligns with standard guidelines for evaluating work based on its scientific contributions rather than the openness of every resource, particularly when sufficient details are provided to ensure reproducibility and understanding.
>
>
>
> ## References:
> [3] SLiC-HF: Sequence Likelihood Calibration with Human Feedback.
>
> ----
> If anything is unclear or requires further elaboration, please don't hesitate to ask, and we would be more than happy to discuss and refine our approach.

---

> > ### Comment · Reviewer_dJek · 2024-11-24
> >
> > Thank you for the response and clarifications.
> > > Unfortunately, we no longer have access to the computational resources required to generate this plot within the rebuttal period...
> >
> >
> > I believe there is no need for additional experiments. All I am asking for is a scatter plot in the style of Figure 1, but with KL on the vertical axis. Figure 1 (left) is very similar to Figure 1 in [1], and it is obvious that the difference between the x-axis (NLL) in this work and the x-axis in [1] should be clearly shown.
> > > In ORPO, KL regularisation is replaced with SFT (using a cross-entropy loss) over preferred completions. This forms the basis for our claim that ORPO can be viewed as a combination of DPO and SFT over preferred completions.
> >
> >
> > Thank you for the clarification. From this, I would like to note one thing: authors are aware of the similarity between SFT loss (NLL(y_w)) and KL divergence.
> >
> > From my perspective, ORPO loss without a regularization term is still very different from DPO loss. According to the results, the dynamics of the metrics depend on the method and should, therefore, be carefully studied.
> > > In response to the reviewer’s comments, we report the L2 loss between the policy model and the reference model with respect to the likelihood.
> >
> >
> > I appreciate the author's effort, but the plot of KL vs NLL is a stumbling block that prevents me from raising my score.

---

> ### Author Response · Authors · 2024-11-25
> **Author Rebuttal by Authors**
>
> We thank the reviewer for engaging in valuable and insightful discussions, particularly over the weekend.
> - **NLL/KL Relationship**. The NLL/KL relationship described by Ravailov's work assumes that completions are sampled according to $\pi_{\text{ref}}$, which is not always the case in our setting. In practice, sampling is often governed by the model’s learned distribution, which may differ from $\pi_{\text{ref}}$.
> - **KL Divergence Dynamics**. It is important to note that the overall KL divergence can only increase during training, as the model’s distribution $\pi_{\text{model}}$ gradually deviates from the reference distribution $\pi_{\text{ref}}$. This is a consequence of learning, where the model adapts to the data and moves away from $\pi_{\text{ref}}$, making direct control of KL divergence challenging.
> - **Challenges in Offline Settings**. As detailed in [1] (Section 4), controlling KL divergence in offline scenarios is inherently difficult. This is because offline loss correlates poorly with actual KL divergence, except in cases where the model distribution remains extremely close to $\pi_{\text{ref}}$. Empirical evidence (e.g., observed order-of-magnitude differences in KL for minimal changes in offline loss) suggests that even small variations in offline loss can lead to significant and unpredictable changes in KL divergence. Given our reliance on fixed datasets without access to online feedback, adjusting KL divergence between $\pi_{\text{model}}$ and $\pi_{\text{ref}}$ in a controlled manner is infeasible in our setup.
>
> While we would like to highlight once again that the relationship between KL and NLL is not central to addressing the research questions our work investigates, and that irrespective of the relationship, observations around scaling effects (particularly with larger models than previously studied) serve to motivate our work and insights. Nonetheless, we agree with the reviewer that the KL vs NLL scatter plot is interesting and will add value to our work. This is a valid and important point; we should have addressed this better in our original paper and included it. This was an oversight on our part.
>
> Due to time and access constraints, we may not be able to generate this plot before the end of the discussion period (although trust us, we are trying), however, we will commit to including the plot in the camera-ready version of the paper.
>
> Once again, we thank the reviewer once again for your engagement and highly valuable contributions.
>
> [1] Generalized Preference Optimization: A Unified Approach to Offline Alignment.

---

> ### Comment · Reviewer_dJek · 2024-11-27
> **Additional thoughts about KL and NLL**
>
> Thank you for the response. Sorry for the pause in discussion. I would like to clarify my thoughts about the NLL vs. KL question.
>
> I found that in [1], the SFT model was used as a starting point for the DAA algorithms; therefore, from this perspective,
> $KL(\pi_\theta \| \pi_{ref})$ and NLL(y_w) are expected to have similar values, because the SFT was learned with the NLL(y_w) objective.
>
> I noticed that you use a slightly different setup (correct me if I'm wrong) with instruction versions of the base models, which were probably trained and maybe aligned on different datasets. In that case, when $\pi_{ref}=\pi_{inst} \neq \pi_{sft}$, your claims may be right.
>
> From this perspective, I would like to emphasize two points:
>
> - Is NLL correlated with KL when $\pi_{ref}$? Yes.
>
> - Is NLL correlated with KL in your case? Probably no.
>
> However, I still think that in [1], it was shown that extreme KL values (in the case where $\pi_{ref} = \pi_{sft}$) lead to suboptimal performance; therefore, you might have the same conclusion that extreme values of NLL(y_w) yield suboptimal performance.
>
> In your case, if you were to train the base model with SFT loss as a first stage, in the second alignment stage with DAA, you would probably see the aforementioned correlation.
>
> > It is important to note that the overall KL divergence can only increase during training.
>
> The likelihood of y_w also tends to decrease (Figure 7 in [1], Figure 2 in [2]) during training, which is a known problem of DPO-like DAAs without an additional NLL(y_w) loss term.
>
> > While we would like to highlight once again that the relationship between KL and NLL is not central to addressing the research questions...
>
> Agreed. However, these results are presented in Figure 1, which is typically used to present the main contribution of the work and therefore draws a lot of the reader's attention.
>
> Additionally, a significant part of the introduction starts with the idea that higher LL(y_w) does not always produce better results, which is presented as a new idea (lines 70-73), but from my perspective, this is not the case.
>
> The second contribution mentioned in the introduction, discussed in lines 80-82 regarding overoptimization signals, is also questionable to me because it is method-dependent.
>
> > Due to time and access constraints, we may not be able to generate this plot before the end of the discussion period.
>
> I understand the authors' position as a person, but as a reviewer, I have to rely on the results of experiments that the authors could have prepared during the discussion period.
>
> Overall, the experiments provided by the authors could be interesting for the alignment community, but the main questions for me are the interpretation and analysis of the obtained results.
>
> [2] [Noise Contrastive Alignment of Language Models with Explicit Rewards](https://arxiv.org/pdf/2402.05369)
>
> If there are any points I may have overlooked or if you would like to discuss any aspect further, please do not hesitate to response.

---

> > ### Author Response · Authors · 2024-11-29
> > **Author Rebuttal by Authors**
> >
> > We sincerely thank the reviewer for their thorough analysis and constructive dialogue throughout this discussion period.
> >
> > > I found that in [1], the SFT model was used as a starting point for the DAA algorithms.
> >
> > We appreciate your analysis of the KL-NLL relationship when $π_\text{ref}$ = $π_\text{sft}$. It makes an excellent point that this relationship has been previously explored in [1], where the reference model was explicitly trained with an NLL objective on preferred completions. Your observation about first training with SFT on preferred completions before DAA alignment is especially insightful - in such cases, we would indeed expect to see the correlation you describe. However, it's important to note that [1]'s setup of using SFT specifically on preferred completions from the preference dataset is actually a special case, not reflective of typical practice.  In most real-world applications, SFT is performed on a separate dataset entirely from the preference data used for alignment. This, combined with our setup using instruction-tuned base models that haven't been explicitly trained to maximize the likelihood of preferred completions, means the KL-NLL relationship is more complex in practical settings.
> >
> > However, we respectfully maintain that our core contributions remain novel and valuable:
> > - Our work is the first to explicitly analyse and quantify the relationship between win probabilities and completion likelihoods across multiple prominent DAA methods (DPO, IPO, SLiC), revealing patterns not previously explored in the literature.
> > - Our framework for detecting overoptimization through combined signals (entropy, top-k mass, etc.) provides practical guidance for practitioners working with various alignment methods.
> > - Our findings about the relationship between win rates and various metrics extend beyond the KL-NLL relationship to provide new insights about optimisation in these methods.
> >
> > The reviewer's feedback has helped us better frame these contributions within existing work, and we are grateful for the opportunity to improve the clarity and positioning of our paper.

---

### Official Review · Reviewer_VwQC · 2024-11-04

**Soundness:** 3
**Presentation:** 4
**Contribution:** 2
**Rating:** 6
**Confidence:** 4

**Summary:**

* The paper studies how completion likelihood affects model performance in Direct Alignment Algorithms (DAAs) like DPO, IPO and Hinge loss, using 7B and 35B models.
* Key finding shows that higher likelihood of better completions and larger margins between better/worse completions don't necessarily improve performance. Higher likelihood helps with factual recall but can hurt diversity.
* Authors propose two metrics for detecting likelihood over-optimization: Decreasing Entropy over Top-k Tokens and Diminishing Top-k Probability Mass. These metrics help prevent over-optimization while maintaining good performance.

**Strengths:**

* The work provides comprehensive experiments across multiple dimensions including likelihood, diversity and performance. The ablation studies are systematic and well-documented.
* The paper challenges common assumptions about likelihood optimization in DAAs. The trade-off between memorization and generalization is clearly demonstrated with empirical evidence.
* The proposed metrics for detecting over-optimization are concrete and actionable. The findings provide clear guidance for practitioners working on DAA training.

**Weaknesses:**

* The paper only studies single epoch training when DPO typically needs 2-3 epochs for best performance. This important limitation is not well justified or analyzed.
* The BINARIZEDPREF dataset lacks crucial details about its construction and preference collection. If preferences come from GPT-4 rather than humans, the generalization of findings is questionable.
* The primary evaluation uses GPT-3.5-turbo as baseline which feels dated. Testing against stronger models like LLaMA-3 would strengthen the findings.

**Questions:**

* Could you provide more information about BINARIZEDPREF construction, including how preferences were collected and validated? The source of preferences is particularly important.
* Why was single-epoch training chosen when DPO typically needs more epochs? Do these patterns persist in multi-epoch training?

---

> ### Author Response · Authors · 2024-11-19
> **Author Rebuttal by Authors**
>
> We appreciate the effort and time taken by the reviewer (VwQC) to provide such a detailed and constructive review. We respond to the main weaknesses and questions below and incorporate much of this in a revised version of the paper, which we believe considerably strengthens our contribution.
>
> > The paper only studies single epoch training when DPO typically needs 2-3 epochs for best performance. This important limitation is not well justified or analyzed.
>
> We chose a single-epoch training setup to focus on understanding the likelihood over-optimisation in DAAs. This approach aligns with prior work [1] demonstrating that over-optimization can manifest in as little as one epoch. While multi-epoch training might yield stronger performance, our findings are specifically targeted at uncovering fundamental dynamics rather than maximising benchmark scores. In future work, we plan to extend our analysis to multi-epoch settings to evaluate the consistency of our conclusions.
>
> [1] Scaling laws for reward model overoptimisation in direct alignment algorithms.
>
> > The BINARIZEDPREF dataset lacks crucial details about its construction and preference collection. If preferences come from GPT-4 rather than humans, the generalization of findings is questionable. Could you provide more information about BINARIZEDPREF construction, including how preferences were collected and validated? The source of preferences is particularly important.
>
> We acknowledge the need for clarity regarding dataset construction. We have provided further information on the construction of BINARIZEDPREF in response to the question below and will add this detail to a revised version of the paper.
>
> The BINARIZEDPREF collection process used a robust multi-source approach combining professional annotators, multiple independent annotation pipelines, and various validation methods. The foundation comes from professional annotation services (~70% of data), with rigorous quality control through multi-annotator consensus, adversarial validation sets, and specialized verification datasets for issues like hallucination and repetition. We've ensured broad domain coverage, incorporating specialised modules for code generation, RAG interactions, STEM, and medical domains while maintaining strong multilingual capabilities across French, Spanish, Korean, Japanese, German, and Italian - including dedicated datasets for handling code-mixing and language transition cases. Quality control is implemented through multiple layers: consensus-based annotation (1-3 annotators depending on complexity), dedicated adversarial validation sets, and specific datasets targeting quality aspects like anti-repetition, length control, and format adherence. The data is predominantly recent (2024), with carefully weighted components and explicit test sets for key capabilities. We use strategic copy multipliers (up to 5x) for crucial capabilities, and the entire dataset is organised into functional groups (multilingual, code, RAG) to ensure balanced training across all target capabilities.
>
> To further validate the generalisation of our findings, we also tested on open-source datasets in our paper, such as UltraFeedback, demonstrating consistent trends and supporting our conclusions.
>
> > The primary evaluation uses GPT-3.5-turbo as a baseline which feels dated. Testing against stronger models like LLaMA-3 would strengthen the findings.
>
> We have already included evaluations against modern LLMs, including GPT-3.5-Turbo, GPT-4o, Claude-3-Sonnet, LLaMA-3-8B, and LLaMA-3-70B-Chat. These results are detailed in Figure 7 of the Appendix. While GPT-3.5-Turbo serves as a baseline, the inclusion of stronger models like LLaMA-3 further validates our findings.
>
>
> While the single-epoch training question is particularly intriguing, we respond to the motivation above and leave exploring multi-epoch training as an extension for future work. We thank the reviewer once again for their detailed and valuable feedback and hope that our revisions have served to clarify the contributions of our work, which we believe will be extremely valuable for informing future research in this direction.

---

### Meta-Review · Area_Chair_GNUz · 2024-12-19

**Metareview:**

This paper studies the relationship between completion likelihood and model performance in Direct Alignment Algorithms (DAAs), analyzing how increased likelihood of better completions and margins between better/worse completions affect performance. The paper claims that higher likelihood doesn't necessarily improve performance and can hurt diversity, proposing two metrics (Decreasing Entropy over Top-k Tokens and Diminishing Top-k Probability Mass) for detecting likelihood over-optimization. The key strengths include comprehensive experiments across multiple dimensions and clear implications for practitioners working on DAA training. However, major weaknesses include: (1) limited novelty, as the observed patterns appear to be consequences of KL divergence effects already documented in prior work; (2) restricted experimental setup using proprietary models/datasets that limits reproducibility; and (3) lack of systematic validation across a broader range of models, datasets and methods. The paper is recommended for rejection primarily because the consensus from the reviewers that it fails to convincingly demonstrate that its findings represent a new phenomenon distinct from known scaling laws of alignment methods.

**Additional Comments On Reviewer Discussion:**

The reviewers have actively engaged in discussions with the authors and held private discussions among the AC/reviewers. During the discussion period, reviewers raised concerns about the relationship between negative log-likelihood and KL divergence, requesting plots to demonstrate their claimed lack of correlation. The authors were unable to provide these plots due to access restrictions. While one reviewer (PsR8) raised their score from 6 to 8 finding value in the systematic experiments, the other reviewers maintained their scores (tR7t:3, dJek:3, VwQC:6), unconvinced by the authors' responses. Key points of contention were: (1) the authors' inability to demonstrate that their findings weren't simply rediscovering known KL divergence effects, (2) limited experimental validation across different models/datasets, and (3) method-dependent nature of the proposed over-optimization signals. The meta-reviewer weighed these concerns, particularly the fundamental question of novelty and reproducibility, more heavily than the value of systematic experiments alone, leading to the reject recommendation.

---

### Decision · Program_Chairs · 2025-01-22

Reject